



# Pre-inversion normal fault geometry controls inversion style and magnitude, Farsund Basin, offshore southern Norway

Thomas B. Phillips[1], Christopher A-L. Jackson[2], James R. Norcliffe[2]

[1]Department of Earth Sciences, Durham University, Science Labs, Durham, DH1 3LE, UK
[2]Basins Research Group (BRG), Imperial College, London, SW7 2BP, UK

*Correspondence to*: Thomas B. Phillips (thomas.b.phillips@durham.ac.uk)

**Abstract.** Inversion may localise along pre-existing structures within the lithosphere, far from the plate boundaries along which the causal stress is greatest. Inversion style and magnitude is expressed in different ways, depending on the geometric and mechanical properties of the pre-existing structure. A three-dimensional approach is thus required to understand how
inversion may be partitioned and expressed along structures in space and time. We here examine how inversion is expressed along the northern margin of the Farsund Basin during Late Cretaceous inversion and Neogene uplift. At the largest scale, strain localises along the lithosphere-scale Sorgenfrei-Tornquist Zone; this is expressed in the upper crust as hangingwall folding, reverse reactivation of the basin-bounding normal fault, and bulk regional uplift. The geometry of the northern margin of the basin varies along-strike, with a normal fault system passing eastward into an unfaulted ramp. Late Cretaceous
compressive stresses, originating from the Alpine Orogeny to the south, selectively reactivated geometrically simple, planar sections of the fault, producing hangingwall anticlines and causing long-wavelength folding of the basin fill. The amplitude of these anticlines decreases upwards due to tightening of pre-existing fault propagation folds at greater depths. In contrast, Neogene shortening is accommodated by long-wavelength folding and regional uplift of the entire basin. Subcrop mapping below a major, Neogene uplift-related unconformity and bore-based compaction analysis show that uplift increases to the north
and east, with the Sorgenfrei-Tornquist Zone representing a hingeline to inversion rather than a focal point, as was the case during the Late Cretaceous. We show how compressional stresses may be accommodated by different inversion mechanisms within structurally complex settings. Furthermore, the prior history of a structure may also influence the mechanism and structural style of inversion that it experiences.

## 1 Introduction

Compressional stresses originating at plate boundaries can be transmitted great distances into continental interiors where they may localise along and drive inversion of pre-existing heterogeneities in the lithosphere (e.g. Berthelsen, 1998; Sandiford and Hand, 1998; Turner and Williams, 2004; Dyksterhuis and Müller, 2008; Buiter et al., 2009; Stephenson et al., 2009; Heron et al., 2018). Such inversion may be expressed via a range of mechanisms, including: i) (ductile) folding of strata (e.g. Jackson et al. 2013; Liboriussen et al., 1987; McClay, 1995); ii) (brittle) reverse reactivation of previously extensional faults (e.g.





Panien et al., 2006; Kelly et al., 2016; Reilly et al., 2017; Patruno et al., 2019; Rodriguez-Salgado et al., 2019; Scisciani et al., 2019); iii) the formation of new reverse and strike-slip faults (e.g. Kelly et al., 1999; Rodriguez-Salgado et al., 2019); and iv) regional basin uplift (Jensen and Schmidt., 1993).

The geometry and mechanical strength of pre-existing structures, and their relative position with respect to adjacent structures, may control if, to what extent, and how these structures will be inverted (Kelly et al., 1999; Walsh et al., 2001; Panien et al.,

2005; Reilly et al., 2017). According to Andersonian fault mechanics, low-angle structures are typically easier to reactivate under compression, whereas sub-vertical structures may be easier to reactivate in a strike-slip sense (Anderson, 1905; Daly et al., 1989). 'Weaker' structures, such as those characterised by high pore pressures and/or pervasive fabrics, are typically easier to reactivate than stronger, more homogeneous structures (Youash, 1969; Gontijo-Pascutti et al., 2010; Chattopadhyay and Chakra, 2013). Furthermore, certain structures may be preferentially reactivated, regardless of their strength, due to their

location with respect to other structures, i.e. those located in the strain shadows of larger structures are unlikely to be reactivated (Walsh et al., 2001; Reilly et al., 2017). Once a structure does reactivate, our understanding of the structural styles and mechanisms of the associated inversion are typically thought of in only a 2D sense, with reverse reactivated faults producing relatively simple inversion-related anticlines (Fig. 1). Such an approach negates the inherent along-strike variability and prior evolution of the inverted structures, and thus leaves a number of unanswered questions. For example, (i) how does

the prior evolution and geometry of a structure affect its propensity to be inverted during compression?; (ii) how does the style and magnitude of inversion vary along-strike?; and (iii) how are different styles of inversion, (i.e. uplift, long-wavelength folding, reverse reactivation etc.) partitioned along-strike of a structure?.

We use borehole-constrained 2D and 3D seismic reflection data from the Farsund Basin, an inverted graben located offshore southern Norway, to examine how compressional stresses were partitioned between different styles and mechanisms of

inversion across a structurally complex rift basin (Fig. 2a). The Farsund Basin has experienced a complex and protracted geological evolution, being located above the westernmost extent of the lithosphere-scale, Sorgenfrei-Tornquist zone (STZ) (Fig. 2a). The STZ corresponds to a pronounced step in lithospheric thickness at sub-crustal depths (e.g. Pegrum, 1984; Mogensen, 1994, 1995; Deeks and Thomas, 1995; Cotte and Pedersen, 2002; Babuška and Plomerová, 2004; Bergerat et al., 2007) and was repeatedly reactivated in response to regional tectonic events (e.g. Carboniferous-Permian transtension, Early

Cretaceous extension, and Late Cretaceous inversion (Fig. 2b) (Mogensen 1995; Berthelsen 1998; Phillips et al. 2018). This multiphase evolution produced a complex rift system along its length, including the Farsund Basin. Further deformation occurred in the Farsund Basin area in the Neogene, due to regional uplift of southern Scandinavia (Japsen and Chalmers, 2000; Baig et al., 2019). This important tectonic event resulted in the formation of a large unconformity and the removal of large thicknesses of strata from across the basin (Fig. 1, 2b).

We focus on the northern margin of the Farsund Basin, which is defined by the complex S-dipping Farsund North Fault in the west and a S-dipping ramp to the east. (Fig. 2a). Specifically, using borehole-based compaction analyses, subcrop mapping, and qualitative (i.e. seismic-stratigraphic) and quantitative analysis of fault-related folds, we examine how inversion was partitioned along this structure throughout Late Cretaceous and Neogene events. Local Reverse reactivation of the basin-





bounding fault occurs locally and was associated with the formation of near-fault hangingwall anticlines and long-wavelength
folding of the basin fill. The hangingwall anticline decreases in amplitude upwards; this is in contrast to that typically
encountered in along inversion-related folds. We suggest that this structural style is related to the presence of fault propagation
folds formed along the fault prior to inversion. We find that reverse reactivation preferentially occurs along geometrically
simple fault sections, areas that display a more complex fault geometry, or where rift-bounding faults are not present, are not
reactivated and inversion is largely manifest as regional uplift. We relate the reverse reactivation of the Farsund North Fault
and the buckling of basin strata to Alpine compressional stresses, localising and buttressed along the lithosphere-scale STZ
and particularly the Farsund North Fault. We attribute the regional uplift and erosion to Neogene uplift of onshore Norway,
with the STZ acting as a hingeline between areas of relative uplift and those of subsidence.

This study highlights how compression, initially localised along the lithospheric-scale STZ, may be expressed via different
mechanisms in the upper crust. We show how inversion-related deformation varies along-strike of a single structure, depending
on the geometric complexity and pre-inversion evolution of the latter.

## 2 Geological setting and evolution

The Farsund Basin, located ~50 km offshore southern Norway, is an E-trending Early Cretaceous graben that underwent
inversion and uplift during the Late Cretaceous and Neogene (Jensen and Schmidt 1993; Mogensen 1995). The basin is defined
by the N-dipping Fjerritslev Fault system along its southern margin and the S-dipping Farsund North Fault along its northern
margin (Phillips et al. 2018). The basin can be separated into an upper and lower terrace by a series of N-S-striking faults that
also define the Varnes Graben, and the Eigerøy and Agder horsts to the north (Fig. 2a). East of the eastern termination of the
Farsund North Fault and the Agder Horst, the northern margin of the Farsund Basin is represented by the S-dipping Agder
Slope (Fig. 2a).

Detailed accounts of the structural and stratigraphic evolution of the Farsund Basin can be found in Phillips et al. (2018) and
Phillips et al. (2019), respectively. We here outline the key pre-inversion tectonics, before detailing how Late Cretaceous and
Neogene inversion events were expressed in the wider region.

### 2.1 Pre-inversion evolution of the Farsund Basin

The Farsund Basin is situated towards the westernmost extent of the Sorgenfrei-Tornquist Zone, which forms the northwestern
section of the lithosphere-scale Tornquist Zone (Pegrum, 1984; Berthelsen 1998; Thybo, 2000; Cotte and Pedersen 2002;
Mazur et al. 2015). The Tornquist Zone represents a sharp change in lithospheric thickness and structure between thick cratonic
lithosphere of the Eastern European Craton to the north and east, and younger, relatively thin lithosphere of central and western
Europe to the southwest (e.g. Kind et al., 1997; Cotte and Pedersen, 2002; Babuška and Plomerová, 2004). At upper crustal
depths, the Tornquist Zone has been periodically reactivated during multiple tectonic events and is described as a 'buffer zone'
to regional tectonic stresses (Mogensen et al., 1995; Berthelsen, 1998).



Carboniferous-Permian transtensional reactivation of the STZ was associated with rift activity and voluminous magmatism, including the emplacement of the WSW-trending Farsund Dyke Swarm (Fig. 2b) (Heeremans and Faleide, 2004; Heeremans et al., 2004; ; Wilson et al., 2004; Phillips et al., 2017; Malehmir et al., 2018). However, no Carboniferous-Permian fault activity is identified in the Farsund Basin, although some likely occurred in the Norwegian-Danish and Egersund basins (Jackson and Lewis, 2013) (Fig. 1, 3).

E-W oriented Triassic extension formed N-S-striking faults, including those that internally dissect the Farsund Basin and those that bound the Varnes Graben (Fig. 2) (Vejbæk, 1990; Phillips et al., 2018). The fault defining the western margin of the Varnes Graben, and those internally dissecting the Farsund Basin, likely formed a single structure during the Triassic (Phillips et al. 2018; 2019). At this time, this structure bounded the eastern margin of the Stavanger Platform (Fig. 2a). During the Triassic, the proto-Farsund Basin resided along the northern margin of the Norwegian-Danish Basin which continued

northwards into the present Varnes Graben (Fig. 2, 3) (Phillips et al. 2019). Upon their deposition, Triassic, and likely Jurassic strata were contiguous with those in the Varnes Graben to the north (Fig. 3).  During the Early-Middle Jurassic, these N-S-striking faults were sinistrally offset by strike-slip activity along E-W-striking faults (Phillips et al. 2018).

Dextral transtensional reactivation of the STZ occurred during the Early Cretaceous (Fig. 2b) (Mogensen, 1995; Erlström et al., 1997; Phillips et al., 2018). This was associated with the formation of the E-W-striking normal faults bounding the Farsund

Basin, forming a graben and separating it from the Norwegian-Danish Basin (Fig. 2, 3). Early Cretaceous extension in the Farsund Basin was associated with relatively rapid slip on the basin-bounding faults and correspondingly high rates of basin subsidence and accommodation generation (Phillips et al., 2018). Sediment accumulation rates eventually outpaced accommodation generation and the faults were buried (Fig. 3). Cretaceous and younger strata are eroded across the study area below the Base Pleistocene Unconformity (Fig. 3).

**2.2 Late Cretaceous and Neogene inversion events**

Late Cretaceous inversion occurred along the length of the Tornquist Zone due to the Alpine-Carpathian orogeny to the south (Fig. 2a) (e.g. Deeks and Thomas, 1995; Berthelsen, 1998; Hansen et al., 2000). West of the Farsund Basin, inversion along the Stavanger Fault System occurred during the Santonian-Cenomanian (Fig. 2a) (Jackson et al., 2013). Alpine inversion is also documented elsewhere in the North Sea (Biddle and Rudolph, 1988; Cartwright, 1989; Jensen and Schmidt, 1993; Jackson

et al., 2013). Uplift and erosion also occurred during the Neogene due to uplift of the South Swedish and South Scandes domes and ridge-push associated with the opening of the North Atlantic (Fig. 2b) (e.g. Jensen and Schmidt, 1993; Japsen and Chalmers, 2000; Japsen et al., 2002, 2018; Kalani et al., 2015; Baig et al., 2019).



## 3 Data and methods

### 3.1 Data

We use a 2D seismic reflection dataset covering the Farsund Basin. This dataset consists of 31 N-trending seismic sections tied by five E-trending sections. These data record to 7 seconds two-way-travel time (s TWT) and are closely-spaced (~3 km), allowing us to correlate our interpretations between individual sections. The seismic data are displayed as zero-phase and follow the SEG reverse polarity convention: a downward increase in acoustic impedance is represented by a trough (red) and a downward decrease in acoustic impedance is represented by a peak (black). The 2D seismic sections data were locally

augmented by a 3D seismic volume providing coverage of the southern margin of the basin, which images to 4 s TWT.

The ages of mapped seismic horizons were constrained by well 11/5-1, located on the southern margin of the basin. Wells 9/3-1, 10/5-1, 10/7-1 and 11/9-1, located outside of the main study area, provided additional age constraints (Fig. 2a). Velocity log information from well 11/5-1, two additional wells within the STZ (J-1, Felicia-1), and four wells in the Norwegian Danish Basin (10/5-1, 10/8-1, F-1, K-1) were used to estimate the amount of uplift across the region (see Sect. 3.4.2). Due to

incomplete coverage of the velocity log through the interval penetrated by 11/5-1, and in order to not introduce additional errors into our measurements, we did not depth convert the data. Well reports were also used to extract lithological information from each well in order to remove unsuitable lithologies from the porosity analyses.

### 3.2 Seismic interpretation

We mapped seven seismic horizons that define the present structure and allow us to constrain the temporal evolution of the

study area (Fig. 3): i) Top crystalline basement, corresponding to the base of a Carboniferous-Permian aged interval; ii) top Acoustic Basement, corresponding to the base Upper Permian Zechstein Supergroup evaporites where present, and the base Triassic where the evaporites are absent; iii) the Base Jurassic Unconformity (BJU); iv) Top Jurassic; v) top Lower Cretaceous; vi) Top Cretaceous; and vii) the Base Pleistocene Unconformity. Additional horizons were interpreted within the Lower Cretaceous interval to help constrain the geometry and evolution of inversion-related structures.

### 145 3.3 Quantitative fault analyses

We calculated throw-length profiles along the Farsund North Fault to determine its kinematic evolution (e.g. Walsh et al., 2003; Duffy et al. 2015; Yielding et al. 2016). To accurately constrain the evolution of a fault we need to account for all slip-related strain, including both brittle faulting and ductile folding. Therefore, where necessary, such as in areas displaying complex fault geometries, or fault-parallel short-wavelength folding, we project horizon cutoffs onto the fault plane from a

regional datum (Fig. 1) (e.g. Walsh et al. 1996; Long & Imber, 2012; Coleman et al. 2018). We minimise potential errors in our measurements by measuring measure fault throw as opposed to displacement, removing the potential for errors associated with depth conversion (Fig. 1). Although small measurement errors inevitably persist in our calculations, these do not affect the overall throw patterns and results. We calculated throw-length plots for the Acoustic Basement, Base Jurassic





Unconformity and Top Jurassic horizons, as these were almost fully preserved in both the hangingwall and footwall of the
Farsund North Fault (Fig. 1). The Top Jurassic horizon is missing in the footwall of the fault in some areas due to erosion
below the Base Pleistocene Unconformity; estimates of throw at this stratigraphic level therefore represents a minimum
estimate. No Triassic strata are preserved on the upper terrace of the Farsund Basin (Fig. 2a); in these areas the Base Jurassic
Unconformity forms a composite surface with the top Acoustic Basement. To quantify the magnitude of inversion experienced
along the Farsund North Fault, we measured the amplitude of the fold at multiple stratigraphic levels. Fold amplitude was
measured from a regional datum unaffected by folding to the fold crest.

### 3.4 Quantifying uplift and erosion

### 3.4.1 Seismic-stratigraphic analysis

To estimate the uplift and erosion that occurred along the northern margin of the Farsund Basin, truncated stratigraphic
horizons were projected above the Base Pleistocene Unconformity. By approximating their pre-erosion stratigraphic thickness,
we can estimate the amount of missing strata and therefore erosion that occurred across the basin (Fig. 4). The top of the
Cromer Knoll Group (Lower Cretaceous) represents the shallowest mapped horizon that is truncated by the Base Pleistocene
Unconformity. This horizon is interpreted as the base of the syn-inversion sequence along the Stavanger Fault System, 100
km to the west (Fig. 2a) (Jackson et al., 2013). The deepest regionally mappable horizon truncated by the Base Pleistocene
Unconformity is typically the Acoustic Basement. We measure uplift between the projections of these horizons, with the
measurement taken at the truncation of the deeper horizon. However, in some areas we measure uplift at the Farsund North
Fault to avoid assumptions about projected stratal thicknesses across the fault (Fig. 4). As a result, our seismic-stratigraphic
technique provides only a minimum estimate of the amount of uplift. We provide two measurements of (minimum) inversion-
related uplift along the basin margin by: (1) projecting strata linearly from the immediate subcrop; and (2) modifying the
subcrop projections to take into account regional thickness changes within the underlying stratigraphic intervals (Fig. 4).

### 3.4.2 Well-based compaction analysis

In addition to using seismic-stratigraphic techniques to estimate inversion-related basin-scale uplift, we also utilise porosity-
depth trends and compaction analyses from seven wells within and surrounding the Farsund Basin (Fig. 2a). This process is
based on the recognition that due to mechanical compaction and diagenesis, sedimentary rocks lose porosity ($\emptyset$) with increasing
burial depth ($z$) (Magara, 1978; Sclater and Christie, 1980). This porosity loss is largely inelastic (Giles et al, 1998), meaning
that if a rock is uplifted it will be overcompacted relative to its new depth of burial (Magara, 1978; Japsen, 1998; Japsen et al.,
2007a). The net magnitude of vertical exhumation ($E_N$) can hence be calculated by subtracting the porosity measured at present-
day burial depth ($B_{present-day}$) from the maximum depth of burial ($B_{max}$) (Corcoran and Doré, 2005):

$$E_N = B_{max} - B_{present-day} \qquad\qquad\qquad (1)$$



$B_{max}$ is commonly predicted from porosity-depth relationships calibrated in settings characterised by continuous subsidence and progressive burial (Jensen and Schmidt, 1993; Doré and Jensen, 1996; Williams et al, 2005; Burns et al, 2007; Japsen et al 2007a; Tassone et al, 2014). Whilst such trends have been generated for discrete lithologies (e.g. Sclater and Christie, 1980), it is more accurate to use regionally calibrated relationships for specific stratigraphic intervals (Japsen et al, 2007b).

Here, we use porosity-depth data to estimate the exhumation of Lower Cretaceous mudstone and to quantify the magnitude of

inversion-related basin-scale uplift. We used the Lower Cretaceous interval because it is the only mudstone-bearing interval encountered in all seven wells, where it varies in vertical thickness from 105-615 m. Six of these wells (10/8-1, 10/5-1, K1, F1, J1 and Felicia-1) were included in previous regional studies of basin exhumation, although different stratigraphic intervals (i.e. Late Cretaceous chalks) were used in these previous analyses (Japsen and Bidstrup, 1999; Japsen et al., 2007). Well 11/5-1, which is located on the south flank of the Farsund Basin (Fig. 2a), has not previously been used for exhumation analysis

and thus provides new constraints on inversion-related regional uplift of the Farsund Basin.

Two porosity-depth relationships have been proposed for Norwegian Shelf Cretaceous-Tertiary shales (Hansen, 1996), one linear and one exponential:

$$\emptyset=0.62-0.00018z \tag{2}$$

$$\emptyset=0.71e^{(-0.00051z)} \tag{3}$$

Although both trends were derived statistically from porosity-depth data from 29 wells from areas that have not been uplifted (Hansen, 1996), we use the exponential relationship because it better conforms to rock physics models in that it does not predict negative porosities at depths greater than 3444 m, as is the case for the linear trend (Japsen et al., 2007b).

To compare well-log data to regional predictions of $B_{max}$, slowness values ($\Delta t$, measured in µs/ft), measured by the sonic log, were converted into porosities using a regionally calibrated modification (Hansen, 1996) of the Wyllie-time average equation (Wyllie et al., 1956):

$$\emptyset=(1/1.57)((\Delta t-59)/(189-59)) \tag{4}$$


Given that this relationship is valid only for shales, intra-Lower Cretaceous sandstone and limestone beds were identified (via the well report) and removed from the analysis. Intra-shale porosities were then averaged over 5 m intervals and plotted against the depth (below the seabed) mid-point for this interval.

Following log editing and porosity estimations, we calculated net exhumation ($E_N$) (Eq. 1) for each porosity-depth datapoint

from each well. These $E_N$ estimates were then averaged to produce one exhumation estimate per well. Across the Farsund Basin, the Base Pleistocene unconformity is overlain by ~100-200 m of sediments (~156 m in 11/5-1) (Fig. 3). This indicates that any exhumation recorded by the Lower Cretaceous shales has been followed by further burial. If not properly accounted





for, this post-uplift burial ($B_E$) will obscure porosity-derived exhumation estimates. The following equation (Corcoran and Doré, 2005) was used to derive the gross exhumation ($E_G$) from the net exhumation ($E_N$):


$$E_G = E_N + B_E \qquad\qquad\qquad\qquad (5)$$

In each well, $B_E$ was assumed to equal the vertical thickness of supra-Base Pleistocene unconformity sediments. Our borehole-based compaction analyses provide spot measurements of uplift in the Farsund Basin and surrounding areas. We combined

these results with those derived from the seismic-stratigraphic techniques described above, allowing us to determine how uplift varied spatially across the area.

## 4 Structural style variability along the northern margin of the Farsund Basin

The structural style of the northern margin of the Farsund Basin varies along-strike (Fig. 5a). Early-Middle Jurassic strike-slip activity resulted in across-fault juxtaposition of different structural elements (i.e the hangingwalls and footwalls of N-S striking

faults), which were then extensionally offset by the Farsund North Fault during the Early Cretaceous (Phillips et al. 2018). Based on the structural elements in the hangingwall and footwall of the fault, we define four structural domains along the northern margin of the basin (Fig. 5a): A) the western end of the Farsund North Fault, with the Eigerøy Horst in the footwall and the upper terrace of the Farsund Basin in the hangingwall; B) a structurally complex zone along the Farsund North Fault with the Varnes Graben in the footwall and the upper terrace of the Farsund Basin in the hangingwall; C) the eastern end of

the Farsund North Fault, with the Varnes Graben in the footwall and the lower terrace of the Farsund Basin in the hangingwall; and D) east of the Farsund North Fault, incorporating the eastern termination of the fault, with the Agder Horst in its footwall, and the Agder slope further east.

### 4.1 Domain A – Reactivation of an older fault?

No Triassic strata are present in Domain A due to erosion at the Base Jurassic Unconformity (Fig. 5a). Throw across the

equivalent Acoustic Basement and Base Jurassic Unconformity horizons is ~1000 ms TWT, whereas the minimum throw across the top Jurassic horizon is ~500 ms TWT; note this is a minimum estimate due to erosion of Jurassic strata across the Eigerøy Horst in the footwall of the Farsund North Fault (Fig. 5a). Earlier, likely Carboniferous-Permian activity may have occurred along the fault in this area although we are unable to determine this due to erosion at the Base Jurassic Unconformity (Fig. 5a). We identify a clinoform bearing interval in the upper Jurassic, likely related to the Farsund Delta system identified

by Phillips et al. (2019). This delta continues northwards into the Varnes Graben, suggesting a relatively young (i.e. post-Late Jurassic and, we suggest, Early Cretaceous) age for uplift of the Eigerøy Horst and corresponding slip on the Farsund North Fault. We identify further clinoform bearing intervals in the Early Cretaceous succession; these may be locally sourced from erosion of the Eigerøy Horst. Lower Cretaceous strata are truncated close to the fault, indicating they have experienced some



uplift. The lack of major hangingwall deformation suggests that the Farsund North Fault in this area experienced little to no
reverse reactivation (Fig. 5a).

## 4.2 Domain B – Complex strike-slip related faulting

Domain B is characterised by a complex zone of faulting consisting of a main central fault that is flanked by numerous
antithetic and synthetic faults (Fig. 5a). Triassic strata are preserved in the footwall (i.e. the Varnes Graben), but eroded from
the hangingwall. In contrast, Jurassic strata are continuous across the fault. Throw is constant (~500 ms TWT) across the fault
for all horizons (Fig. 5b). Cretaceous strata thicken from the footwall to hangingwall of the fault and are truncated at the
overlying Base Pleistocene Unconformity. Phillips et al. (2018) estimate this domain experienced ~10 km of sinistral strike-
slip offset during the Early-Middle Jurassic. The proposed strike-slip fault continues towards Domain A to the west, and
continues to the southeast, south of Domain C, to the east. Based on the truncation of Lower Cretaceous strata, inversion may
have caused uplift and compression and uplift of Domain B, although this is difficult to determine.

## 4.3 Domain C – Reverse reactivation and hangingwall folding

Triassic and Jurassic strata are isochronous and display relatively constant throw across Domain C (Fig. 5a, b). Throw increases
from ~600 ms TWT in the west to a maximum of ~1100ms TWT in the centre, before decreasing to ~800 ms TWT at the
boundary with the Agder Horst to the east (Fig. 5b). Unlike Domain B, this area did not experience any strike-slip faulting,
with the proposed strike-slip fault continuing to the southeast, south of the Farsund North Fault in this area (Phillips et al.,
2018). The Farsund North Fault in this area is represented by a straight structure that slipped only during the Early Cretaceous
(Fig. 5a). Some antithetic faults in the hangingwall merge with the main structure at depth. A large anticline is present in the
hangingwall of the Farsund North Fault in Domain C, the amplitude of which is greatest at the top of the Jurassic and decreases
towards shallower levels (Fig. 5c). The amplitude of the fold also decreases towards the edges of Domain C, suggesting it has
a periclinal geometry. Lower Cretaceous strata are truncated by the Base Pleistocene Unconformity and are not preserved on
the footwall of the fault. We suggest that Domain C represents an Early Cretaceous segment of the Farsund North Fault, which
propagated away from a pre-existing segment (Domain A) during the Early Cretaceous, with Domain B situated between the
two segments.

## 4.4 Domain D – Basin uplift and erosion

Domain D is characterised by the S-dipping Agder Horst, which hosts numerous low-displacement, E-W-striking, N-dipping
faults. The Farsund North Fault terminates eastwards into numerous S-dipping splays, potentially due to the presence of the
Farsund Dyke Swarm within basement (Fig. 5a) (Phillips et al., 2017). Strata are progressively truncated northwards at the
Base Pleistocene Unconformity, indicating wholesale basin-scale uplift and erosion (Fig. 5a).





# 5 Styles of inversion

## 5.1 Regional uplift and erosion

Strata along the northern margin of the Farsund Basin are variably truncated at the Base Pleistocene Unconformity. We here use compaction analyses and seismic-stratigraphic methods to quantify the inversion-related uplift driving this erosion.

### 5.1.1 Well-based compaction analyses

We plotted porosity-depth data from seven wells against the Lower Cretaceous compaction curve of Hansen (1996). This normal compaction curve, which assumes continuous burial and hydrostatic stress conditions, lies below data from each all of

the wells (Fig. 6a), indicating that overcompaction is present regionally.

Porosity-depth data cluster in two depth-intervals (170-825 m and 1080-1460 m depth; Fig. 6a). The shallower cluster contains data from wells 11/5-1, Felicia-1 and J1, all of which are located within the STZ (Fig. 2a). These wells record gross-exhumation values 647-775 m, with 11/5-1 recording the greatest amount of exhumation (775 m) (Fig. 6b). The deeper cluster contains data from wells 10/5-1, 10/8-1, F-1 and K-1; all of which are situated south of the Farsund Basin and STZ (Fig. 2a). Exhumation

estimates from these wells are overall lower than those to the north, varying from 273-683 m. Well 10/5-1, which is located closest to the STZ, records the highest gross exhumation (683 m), whereas wells 10/8-1, F-1 and K-1 record values of 270 m, 405 m, and 295 m respectively (Fig. 6b). In summary, our analyses show that the amount of exhumation increases towards the STZ (Fig. 2a); no well information is available further north.

### 5.1.2 Seismic-stratigraphic analyses

By projecting strata as a straight line based on the dip at the point they are truncated beneath the Base Pleistocene Unconformity, we estimate ~100 ms TWT uplift in Domain A, increasing to a maximum of ~400 ms TWT in Domain C. this value then decreases to ~250 ms TWT across the Agder Slope (Fig. 5c). Our other technique which varies the dip of the projected strata to account for underlying stratal geometries, suggests ~200 ms TWT of uplift in Domain A increasing to ~400 ms TWT in Domains B and C these values are thus similar to those obtained using a straight projection method. Corrected

projection measurements predict ~800 ms TWT in Domain D (Fig. 5c). Here, the straight projection method underestimates uplift as truncated strata become more steeply south-dipping to the north. This suggests an increase in uplift to the north.

Across the Farsund Basin, uplift increases eastwards, particularly past the eastern termination of the Farsund North Fault (Domain D). No reverse reactivation of the fault occurs in this area (see below) and bulk, regional uplift appears to represent the main mechanism accommodating inversion-related shortening (Fig. 7). Older strata progressively subcrop the Base

Pleistocene Unconformity towards the north and east (Fig. 7, 8). Acoustic Basement strata subcrop the Base Pleistocene Unconformity on the Eigerøy and Agder horsts, due to their location in the footwalls of the bounding faults of the Varnes Graben. Deep strata also subcrop the Base Pleistocene Unconformity further east along the Agder Slope. Jurassic and Lower Cretaceous strata subcrop the Base Pleistocene Unconformity in the Varnes Graben (Fig. 8).





## 5.2 Evidence for fault reactivation

A prominent hangingwall anticline occurs in Domain C of the Farsund North Fault (Fig. 5a, 9). The fold incorporates Upper Triassic to Lower Cretaceous strata, before it is truncated upwards by the Base Pleistocene Unconformity (Fig. 9). The fold is ~35 km long, extending from the westernmost part of Domain B in the west and terminating to the east next to the Agder Horst in Domain D (Fig. 5a, c). The amplitude of the fold is greatest in the middle of Domain C (~200 ms TWT), decreasing to zero at its lateral terminations. The amplitude of the fold varies with depth; it is similar at Top Jurassic and Base Jurassic

Unconformity levels (~150 ms TWT), decreasing upwards into the Lower Cretaceous (~80 ms TWT) (Fig. 5c). The fold is typically tightest close to or just above the top of the Jurassic interval, where it is often deformed by normal faults along its hinge (Fig. 9a). Lower Cretaceous strata thin by up to 50% onto the hangingwall-facing limb of the fold, and often onlap onto this limb at deeper levels (Fig. 9b, c).

Stratigraphic thinning and onlap onto fault parallel hangingwall folds are characteristic of extensional growth folds (also known

as 'fault-propagation folds' or 'forced folds'; see review by Coleman et al., 2019). These folds, which are initially expressed as basinward-facing monoclines, typically form above the propagating upper tiplines of blind normal faults (Mitra and Islam, 1994). These folds are subsequently breached during subsequent fault slip and tip propagation. However, in the Farsund Basin, the fault-parallel fold is anticlinal as opposed to monoclinal, suggesting that this is not the cause of the folding. Anticlinal fault-parallel hangingwall folds may form as fault-bend folds due to changes in fault dip (e.g. Suppe, 1983; Withjack and

Schlische, 2006). However, we discount such an origin here due to the relatively planarity of the fault and lack of major dip changes (Fig. 9).

Based on the observations outlined above, we interpret the fault-parallel hangingwall fold as an inversion-related anticline (e.g. Dart et al., 1995; Lowell, 1995; Turner and Williams, 2004; Yamada and McClay, 2004). Due to erosion at the Base Pleistocene Unconformity, the crest of the fold and any associated growth strata are not preserved, providing no direct

constraints on the timing of fold formation or causal inversion (Fig. 1, 9). However, the folding (and inversion) must have occurred post-Early Cretaceous, given Lower Cretaceous strata are incorporated in the fold (Fig. 9).

We also identify a minor graben in the centre of the Farsund Basin, bound by faults that span a depth range of ~1000 ms TWT in the thickest part of the Lower Cretaceous syn-rift succession (Fig. 10). These faults are basement detached, terminating downwards at ~1500 ms TWT and upwards, either within the upper levels of the Lower Cretaceous interval or by truncation

at the Base Pleistocene Unconformity (Fig. 10). They are associated with relatively low displacements (~10-20 ms TWT) that are greater at shallower levels (~20 ms TWT). At depth, fault displacements are typically below seismic resolution but the faults can be identified by clear fault plane reflections (Fig. 10).

We interpret these structures as outer-arc flexural faults formed in response to basin compression and related hangingwall buckling (e.g. Panien et al., 2005, 2006). The decrease in throw with depth suggests arching of the basin fill, whilst their

relatively uniform termination depth may define the fold neutral surface, with areas above under extension, and below under overall compression (Fig. 10). A ~65 ms TWT thick clinoform-bearing interval is present at ~750 ms TWT in the Lower



Cretaceous interval, prograding basinwards from the southern margin of the basin (Fig. 10). The presence of basinwards prograding clinoforms indicate that the basin still represented a depocentre at that time.

## 6 Discussion

### 6.1 Structural styles and expression of inversion

Inversion-induced hangingwall folding only occurs locally along the Farsund North Fault (Domain C), with little direct evidence for reverse reactivation being observed elsewhere (Fig. 5a). Domain B is characterised by a complex zone of faulting formed during Early-Middle Jurassic strike-slip faulting (Phillips et al., 2018); this domain is located between two segments of the Farsund North Fault of potentially differing ages (Fig. 5a). The eastern fault segment (Domain C) only initiated in the Early Cretaceous, with Carboniferous-Permian strata being isopachous across the fault (Fig. 3). The western segment of the Farsund North Fault was also active during Early Cretaceous extension, and may have been active earlier during Carboniferous-Permian extension, although we are unable to confirm this due to a lack of preserved strata (Fig. 5a). Similarly, along the southern margin of the Farsund Basin, the southern strand of the Fjerritslev Fault System was inactive prior to the Early Cretaceous extension, at which time it propagated westwards from a pre-existing segment of the fault (Fig. 2a) (Phillips et al., 2018). We also suggest that the Farsund North Fault propagated eastwards in the Early Cretaceous from a pre-existing fault segment (Domain A). The eastern segment thus represents the youngest, and accordingly the least complex, section of the Farsund North Fault, forming a simple planar structure (Fig. 9). We suggest that the relatively simple geometry of the eastern segment of the Farsund North Fault caused it to be preferentially reactivated during Late Cretaceous inversion. Whilst other sections of the fault may have been weaker, they were not reactivated due to their more complex geometry, related to their prior evolution and segmentation. Similar links are made between highly deformed and complex fabrics within shear zones (such as those typified by recumbent and isoclinal folding) and the lack of major fault exploitation during subsequent extension as the complexity of the fabric blocks lateral fault propagation (Morley, 1995; Salomon et al., 2015). Along the Farsund North Fault, local stress field interactions between different structures and fault segments may result in an obscuring of any pervasive anisotropy, inhibiting strain localisation and therefore reactivation.

Typically, during reverse reactivation, pre-inversion strata in the hangingwall of the fault are folded into an inversion-related anticline of constant amplitude with depth (Fig. 1) (Mitra and Islam, 1994; Lowell, 1995; McClay, 1995). However, observations of the eastern segment of the Farsund North Fault (Domain C) diverge from this, with fold amplitude decreasing towards shallow depths (Fig. 5c, 9).

Some of this decrease in amplitude may be attributed to the interplay between the folding of competent and incompetent units in the core of the fold. More competent units will deform via folding and maintain a constant thickness, whereas less competent units will not maintain thickness and may be extruded from the core of the fold. The increased compaction of deeper buried strata would result in a potential decrease in fold amplitude with depth, the opposite to that observed in the Farsund Basin (Fig. 5c). The Lower Cretaceous interval comprises relatively homogeneous siltstones, with no major changes in lithology and





competency expected. Although lithologies do vary within the Jurassic interval (Phillips et al., 2019), this does not correlate
to the location of the change in fold amplitude (Fig. 9). Furthermore, the magnitude of the amplitude change is likely too large
to be explained by the mechanical properties of the strata alone.

Along the Farsund North Fault, strata within the lower sections of the Lower Cretaceous interval onlap folded strata within the
inversion anticline and thin onto the hangingwall limb of the fold (Fig. 9). These stratal relationships suggest some relief at
the free surface in the hangingwall of the Farsund North Fault during Early Cretaceous extension and fault slip. Fault
propagation folding of Jurassic and Early Cretaceous strata occurs along the southern margin of the Farsund Basin, associated
with Early Cretaceous faulting (Phillips et al., 2018); we suggest that the Farsund North Fault may have experienced similar
fault-propagation folding during Early Cretaceous extension.

Based on the variable fold amplitude and stratal relationships within the hangingwall of the Farsund North Fault, we propose
the following model to explain its inversion and the resultant structural style. The eastern segment of the Farsund North Fault
(Domain C) formed as a new structure, propagating eastwards during Early Cretaceous extension (Fig. 2a, 5a). This extension
was associated with fault propagation folding of Jurassic and lowermost Lower Cretaceous strata, which was subsequently
breached and buried by the uppermost Lower Cretaceous succession (Fig. 11a, b). During inversion, folding of near-fault strata
within the upper parts of the Lower Cretaceous succession produced a ~60 ms TWT amplitude hangingwall anticline (Fig. 5c,
12c). However, at deeper structural levels, where pre-inversion strata were already folded, the initial monoclinal fold was
tightened and rotated, forming an inversion-related anticline (Fig. 12c). The amplitude of the resultant composite fold thus
reflects both the Early Cretaceous fault propagation folding and subsequent inversion, similar to those produced by Mitra and
Islam (1994). Towards the lateral terminations of the fold, at the boundaries of Domain C, we observe a more monoclinal fold
geometry (Fig. 9c), as these areas experienced less inversion and therefore preserve the initial monoclinal fold geometry.

## 6.2 Temporal and spatial partitioning of inversion styles

The Farsund Basin experienced at least two phases of inversion during the Late Cretaceous and the Neogene (Hansen et al.,
2000; Gemmer et al., 2002; Kalani et al., 2015). During these events, the deformation mechanisms accommodating inversion
were spatially partitioned; regional uplift occurred along the northern margin of the basin (Fig. 7), the Farsund North Fault
underwent local reverse reactivation (Fig. 9), whereas the basin fill was buckled into an open long-wavelength anticline (Fig.
10). Due to truncation of syn-inversion strata at the Base Pleistocene Unconformity, we cannot directly assign these different
inversion mechanisms to the Late Cretaceous or Neogene events.

Late Cretaceous inversion is related to the Alpine-Carpathian Orogeny, with three distinct tectonic pulses recognised in the
Danish area during the Late Cretaceous and Paleogene (Hansen et al., 2000; Gemmer et al., 2002). Within the upper crust, the
STZ is defined by a zone of Late Cretaceous inversion (e.g. Pegrum, 1984; Liboriussen et al., 1987; Michelsen and Nielsen,
1993; Mogensen and Jensen, 1994; Deeks and Thomas, 1995; Mogensen, 1995; Lamarche et al., 2003; Bergerat et al., 2007).
Along the Tornquist Zone, Late Cretaceous inversion is expressed via numerous mechanisms, including reverse fault
reactivation (Kryzwiec, 2002), transpression (Deeks and Thomas, 1995), and basin-scale compression and uplift (Liboriussen





et al., 1987; Erlström et al., 1997; Hansen et al., 2000). Late Cretaceous inversion has also been observed westwards along-strike of the STZ in the Egersund Basin (Pegrum, 1984; Sørensen et al., 1992; Phillips et al., 2016), where inversion along the Stavanger Fault System initiated during the Coniacian (~86.3 Ma) to Santonian (~82.6 Ma) and continued until the

Maastrichtian (~66 Ma) (Jackson et al. 2013). Inversion was relatively mild in the Egersund Basin, producing inversion-related folds of 300-450 m amplitude (Jackson et al., 2013). In comparison, the degree of inversion accommodated by hangingwall folding within the Farsund Basin produces an inversion anticline that has an amplitude of ~80 ms TWT at shallower depths. We suggest the amplitude of the fold at shallow depths is more representative of the structural style forming during Late Cretaceous-Neogene inversion, with the amplitude of the fold at greater depths accentuated by earlier fault propagation folding

(Fig. 11, see above). Some components of inversion within the Farsund Basin may also be partitioned into regional uplift and long-wavelength folding of the basin fill, as well as the reverse reactivation of the previously extensional normal faults and associated hangingwall folding.

Based on similarities in the magnitude and style of inversion between the Farsund and Egersund basins, we suggest that the reverse reactivation of the Farsund North Fault likely occurred during the Late Cretaceous. Long-wavelength folding of the

basin fill and formation of outer-arc flexural faults resembles long-wavelength folding and the formation of Tornquist Zone-adjacent Late Cretaceous depocentres observed elsewhere (Fig. 10) (Liboriussen et al., 1987; Japsen et al., 2002), also suggesting this deformation is Late Cretaceous. At this time, the Farsund North Fault acted as a buttress to Late Cretaceous compression, undergoing reverse reactivation and folding in the immediate hangingwall with simultaneous long-wavelength folding and uplift of the basin fill (Fig. 12a).

As well as reverse fault reactivation and long-wavelength folding of the basin fill, the entire northern margin of the Farsund Basin was uplifted, as is evident from subcrop projections (Fig. 7). In contrast to Late Cretaceous compression, this uplift is not buttressed by the Farsund North Fault, increasing north and east towards the Norwegian mainland and across the Agder Slope (Fig. 5c). Some of this uplift may be attributed to Late Cretaceous compression, although we are unable to distinguish between the two events due to a lack of preserved post-Late Cretaceous strata (Fig. 7). Strata are tilted and truncated along the

eastern margin of the North Sea rift towards the Norwegian mainland, related to the onset of North Atlantic rifting and potential magmatic underplating (Japsen, 1998; Japsen and Chalmers, 2000; Gemmer et al., 2002; Baig et al., 2019). Offshore southern Norway, Neogene uplift is specifically related to the uplift of the South Scandes Dome beneath southern Norway to the north, exposing Proterozoic basement onshore, and uplift of the South Swedish Dome beneath Southern Scandinavia to the east (Jensen and Schmidt, 1992; Japsen et al., 2002). The distribution of these domes is consistent with observations of uplift from

the Egersund Basin (Kalani et al., 2015) and our observations of the increase in uplift to the north and east being related to the South Scandes and South Swedish domes respectively.

### 6.3 Localisation of inversion along pre-existing structures

The STZ may act as a weak buffer zone to regional tectonic stresses, effectively shielding the cratonic lithosphere of the Eastern European Craton to the north and east (e.g. Berthelsen, 1998; Hansen et al., 2000; Mogensen and Korstgård 2003).



During Late Cretaceous inversion, the pronounced change in lithospheric thickness and properties across the zone localises far-field stresses originating from the south associated with the Alpine Orogeny (Fig. 12a). Such long-lived and lithosphere-scale structures, upon which strain can localise, are lacking in the relatively young lithosphere beneath Central and Western Europe and the North Sea (Pharaoh, 1999).

Numerical modelling has highlighted that deformation will localise along weak lithosphere-scale structures, such as the STZ
(Gemmer et al., 2002). Far-field stresses focus along heterogeneities situated at great depths or spanning large depth ranges within the lithosphere. Structures within the mantle lithosphere can control the location of rifting in extensional settings (i.e. the Davis Strait in the Labrador Sea) and orogenic belts in compressional settings (i.e. the Ouachita Orogeny, USA) (Heron et al., 2018, 2019). Rift basins may also form crustal-scale heterogeneities prone to inversion, even long after extension stops. Weaknesses developed at mid-crustal depths during rift-related crustal necking, or irregularities along the Moho, may prime
rift systems for later inversion (Hansen and Nielsen, 2003; Buiter et al., 2009). Crustal-scale faults, such as those bounding the Farsund Basin, also weaken the lithosphere and increase the likelihood of inversion during subsequent regional compression (Lie and Husebye, 1994; Hansen and Nielsen, 2003; Phillips et al., 2018).

At the crustal scale, inversion preferentially occurs on larger structures that are typically weaker, having experienced more deformation (Reilly et al., 2017). The geometry and relative location of fault segments and systems within a larger fault array,
as opposed to the inherent strength of the fault itself, forms a primary control on whether, and to what extent a structure will reactivate (Walsh et al., 2001; Reilly et al. 2017). Optimally-located structures tend to increasingly localise strain, growing larger at the expense of smaller structures residing in 'strain shadows', which ultimately become inactive. The larger, typically more continuous structures are preferentially reactivated in later events. Although areas of the Farsund North Fault may have experienced more deformation and be weaker than other areas, they often display a more complex geometry than newly formed
structures. For example, the eastern segment of the Farsund North Fault represented a single structure that formed a focal point for Late Cretaceous inversion; faults bounding the southern margin of the Farsund Basin were not inverted. The southwards dip of the Farsund North Fault, coupled with its crustal-scale geometry and location along the northern boundary of the STZ buttress, mean the fault was ideally situated to accommodate Late Cretaceous inversion-related compressional stresses (Fig. 12a).

Whilst the STZ and the Farsund North Fault localise stresses during Late Cretaceous inversion, this is not apparent during Neogene uplift (Fig. 12). Uplift increases regionally from zero in the Norwegian-Danish Basin south of the study area, to 1000 m across the Skagerrak-Kattegat Platform to the northeast (Japsen et al., 2002). Based on basin modelling and regional borehole-based porosity analyses of Upper Cretaceous chalks, Japsen et al., (2002) propose 600-800m uplift along the STZ, consistent with our estimation of 775m uplift in the Farsund Basin, based on borehole-based porosity analysis of the Lower
Cretacoeus interval penetrated in well 11/5-1. These values are also consistent with our estimates of uplift from the Felicia-1 (640 m) and J-1 (655 m) boreholes located further eastwards along the STZ. Uplift decreases southwards away from the STZ to around ~300 m within the Norwegian-Danish Basin. Relatively high uplift values (~400m) are documented in the F-1 well (Fig. 2a), although this may in part relate to local salt mobilisation (Fig. 3). The regional south-north uplift gradient drastically





increases at the STZ, as evidenced by the two clusters of uplift estimates from nearby wells; those wells situated atop the STZ,
typically experience 600-800m uplift whereas those further the south document only ~300m (Fig. 2a, 6a). Uplift within the
STZ may be locally augmented by Late Cretaceous uplift related to fault reactivation and localised hangingwall uplift;
however, as the uplift increases away from the STZ and the Farsund North Fault to the north and east, we suggest that any pre-
Neogene (i.e. Late Cretaceous) uplift contributes only a negligible amount of uplift to the regional values. Locally, within the
Farsund Basin, uplift increases northwards away from the STZ (Fig. 12b). To the east, where uplift also increases, the STZ
rotates to strike NW-SE, such that east of Domain D may be located north of the STZ and uplift again increases. Rather than
localising deformation as it does during Late Cretaceous inversion, we suggest that the STZ represents a relative hingeline to
uplift during the Neogene (Fig. 12b), supporting the interpretation of previous regional studies (Japsen et al., 2007; 2018)

## 7 Conclusions

In this study, we show how inversion is expressed along a complex pre-existing structure via various different mechanisms
throughout Late Cretaceous and Neogene compressional events.

Late Cretaceous compression was accommodated via selective reverse reactivation of the Farsund North Fault, forming a
prominent hangingwall anticline. This reactivation occurred along the relatively young and geometrically simple eastern
segment of the fault, which propagated from a pre-existing structure during Early Cretaceous extension. We suggest that the
likelihood of a structure to be reactivated and undergo inversion is not solely related to the size and 'weakness' of the structure;
the relative complexity of the structure also plays an important role. We find that the geometrically simple areas of the Farsund
North Fault are preferentially inverted, whereas those with a more complex geometry, likely having experienced a more
protracted evolution, typically do not localise strain and are not inverted. Late Cretaceous compression was also expressed as
long-wavelength folding of the basin fill, buttressed against the Farsund North Fault, resulting in basin-scale uplift and erosion.
We find that the geometry and prior evolution of a structure also influences the structural style of the resultant inversion-related
structures. The presence of an extensional fault propagation fold, formed prior to inversion, along the Farsund North Fault
results in the formation of a hangingwall anticline that decreases in amplitude upwards during inversion. Fold amplitude at
shallow depths reflects the inversion event, whereas fold amplitude at depth reflects both inversion-related and earlier fault
propagation folding.

During the Neogene, the Farsund Basin experienced widespread uplift related to uplift onshore Norway and Sweden. Based
on borehole-based porosity analyses within the Farsund Basin and surrounding area, we find that uplift increased to the north
and east of the basin, with relatively smaller amounts of uplift occurring in the Norwegian-Danish Basin to the south. Where
the Farsund North Fault was not present, this represents the only mechanism of inversion expressed in the basin.

Late Cretaceous compression localised along the lithosphere-scale STZ; in contrast, Neogene stresses were not localised along
the STZ, with the STZ appearing to represent a relative hinge-line during this time, separating areas of low uplift to the south,
from relatively large amounts of uplift further north.





The Farsund North Fault along the northern margin of the STZ acted as a buttress to compression within the upper crust, with inversion expressed as reverse reactivation, long-wavelength hangingwall folding and regional basin-scale uplift. At upper crustal depths, the prior evolution and geometric complexity of a pre-existing structure plays an important role in how and to what extent that structure may be reactivated during late compression.


*Data availability:* The seismic data used throughout this study are publically available for download via the DISKOS online portal (https://portal.diskos.cgg.com).


*Author Contributions:* TP is the primary author on this study, responsible for the interpretation and writing of the manuscript. CALJ contributed to the genesis of the manuscript and the writing and editing of the manuscript. JN carried out to the borehole-based compaction analyses and contributed to the writing and editing of the manuscript.

*Competing interests:* The authors declare that there are no competing interests.

*Acknowledgements:* This research is funded by a Leverhulme Trust Early Career Fellowship awarded to Phillips. The authors would also like to thank Schlumberger for providing academic licences to Durham University and Imperial College for use of the Petrel Software.

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



**Figure 1: Conceptual model of the reverse reactivation of a previously extensional normal fault. Ductile folding occurs in the hangingwall of the fault producing an anticline displaying a constant amplitude with depth. Stata onlapping onto the limb of the fold at the free surface indicate the age of folding and therefore inversion. In the case of the Farsund Basin, the upper part of the anticline is eroded by the Base Pleistocene Unconformity, meaning that we have no constraints on the timing of inversion. Projected hangingwall cutoffs and the footwall cutoff used for throw analyses are also shown.**

**Figure 2: A) Regional two-way-time (TWT) structure map showing the top Acoustic Basement surface (typically base Triassic/base Zechstein Supergroup evaporites) across the region. The locations and calculated uplift values of wells, used to constrain the ages of stratigraphic horizons across the area are shown. Inset - Regional map showing the location of the study area and Sorgenfrei-Tornquist Zone. B) Tectonostratigraphic column for the Farsund Basin based on lithological information from well 11/5-1 (after Phillips et al., 2019), showing the main lithologies and tectonic events.**

**Figure 3: A) Interpreted N-S oriented seismic section highlighting the present geometry of the Farsund Basin. B) The same section flattened on the Base Jurassic Unconformity highlighting the rough structural and stratal geometries present at the beginning of the Jurassic. The Farsund Basin has not formed at this time and is contiguous with the Varnes Graben and Norwegian-Danish to the north and south, respectively. Triassic strata show only regional thinning towards the north, with salt mobilisation occurring down the paleoslope into the Norwegian-Danish Basin. See Figure 2 for location.**

**Figure 4: Schematic diagram showing how uplift is calculated by projecting truncated stratigraphy above an unconformity. Uplift estimates based on straight and modified projections of strata are calculated. Where possible, measurements are taken at the largest possible value, however, projections are not calculated across faults.**

**Figure 5: A) Structural map and seismic sections highlighting the characteristic structural style associated with each fault domain (A-D). Black lines show the location of individual domain sections. B) Throw-length profiles calculated across the Farsund North Fault for the Acoustic Basement, Base Jurassic Unconformity and Top Jurassic horizons. Background colours correspond to different domains. C) Calculated uplift values (red and black) calculated along the northern margin of the Farsund Basin through the projections of truncated strata. Blue lines show the amplitudes of the hangingwall inversion folds at various structural levels.**

**Figure 6: A) Lower Cretaceous porosity-depth data from seven wells in the region, including one from the Farsund Basin (11/5-1). Depth measured in metres below seabed (m bSB). A normal compaction curve for Norwegian Shelf Lower Cretaceous shales is also shown (Hansen, 1996a). B) Estimates of net exhumation ($E_N$) and gross exhumation ($E_G$), which takes into account the vertical thickness of supra-Pleistocene unconformity sediments ($B_E$). Also shown are values of $E_G$ from earlier studies using different stratigraphic intervals from several of the same boreholes. The difference between these estimates of $E_G$ and those calculated in this study are shown ($E_g$ difference). See Figure 2 for the locations of each of the wells.**

**Figure 7: Interpreted seismic section across the Farsund Basin and Agder Slope, showing subcrop projections, taking into account underlying stratal geometries, and associated uplift. See Figure 2 for location.**

**Figure 8: Subcrop map across the Farsund Basin at the Base Pleistocene Unconformity. Black lines show the locations of faults that are truncated by the Base Pleistocene Unconformity, thick grey lines show fault geometries at the Acoustic Basement level. Thin grey lines show the location of 2D seismic sections used to create the map, with those referred to elsewhere in the study in red.**

**Figure 9: Uninterpreted and interpreted seismic sections highlighting the along-strike variability in fold geometry within Domain C. See Figure 5a for locations. A) Fold geometry in the west of Domain C showing a hangingwall anticline which decreases in amplitude upwards. B) Section from the centre of Domain C, note how lowermost Lower Cretaceous strata onlap onto the limb of the fold at deeper levels. C) Section from the east of Domain C, note how Lower Cretaceous strata thin onto the limb of the fold at deeper levels.**

**Figure 10: Interpreted seismic section across the centre of the Farsund Basin. See Figure 2 for location. Lower and Upper Cretaceous strata are truncated at the Base Pleistocene Unconformity and a series of low-displacement faults are developed in the centre of the basin.**

**Figure 11: Schematic model illustrating the inversion of a pre-existing fault propagation fold. A) An initial stage of fault propagation folding occurred during Early Cretaceous slip along the fault. Lower Cretaceous strata onlap onto the basinward-facing limb of the fold. B) Following further slip along the fault, the fault propagation fold becomes breached. C) Reverse reactivation of the fault and**





folding of hangingwall strata occurs during the Late Cretaceous. At shallow depths, this creates an inversion-related monocline, whereas at greater depths, the pre-existing monocline is tightened, forming a composite Early Cretaceous fault propagation and
750    Late Cretaceous inversion-related fold.

Figure 12: The different styles by which compressional stresses are expressed along the Sorgenfrei-Tornquist Zone throughout Late Cretaceous and Neogene events. A) The STZ localises compression originating from the south, at upper crustal depths, this is buttressed against the Farsund North Fault via reverse reactivation and long-wavelength folding of hangingwall strata. Right - the STZ acts as a hinge line during the Neogene, separating areas of relatively low uplift in the Norwegian-Danish Basin, from the areas
755    of relatively high uplift further north towards the Norwegian mainland.



Normal faulting

Strata not preserved
in Farsund Basin

Onlapping onto
inversion anticline

Inversion fold
amplitude

FW cutoff

Inversion
anticline

Truncation of strata
at Base Pleistocene
Unconformity

Displacement

Throw

Heave

HW projection

HW cutoff

Reverse fault
reactivation

Reverse reactivation
and regional uplift

# Figure 1



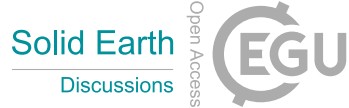

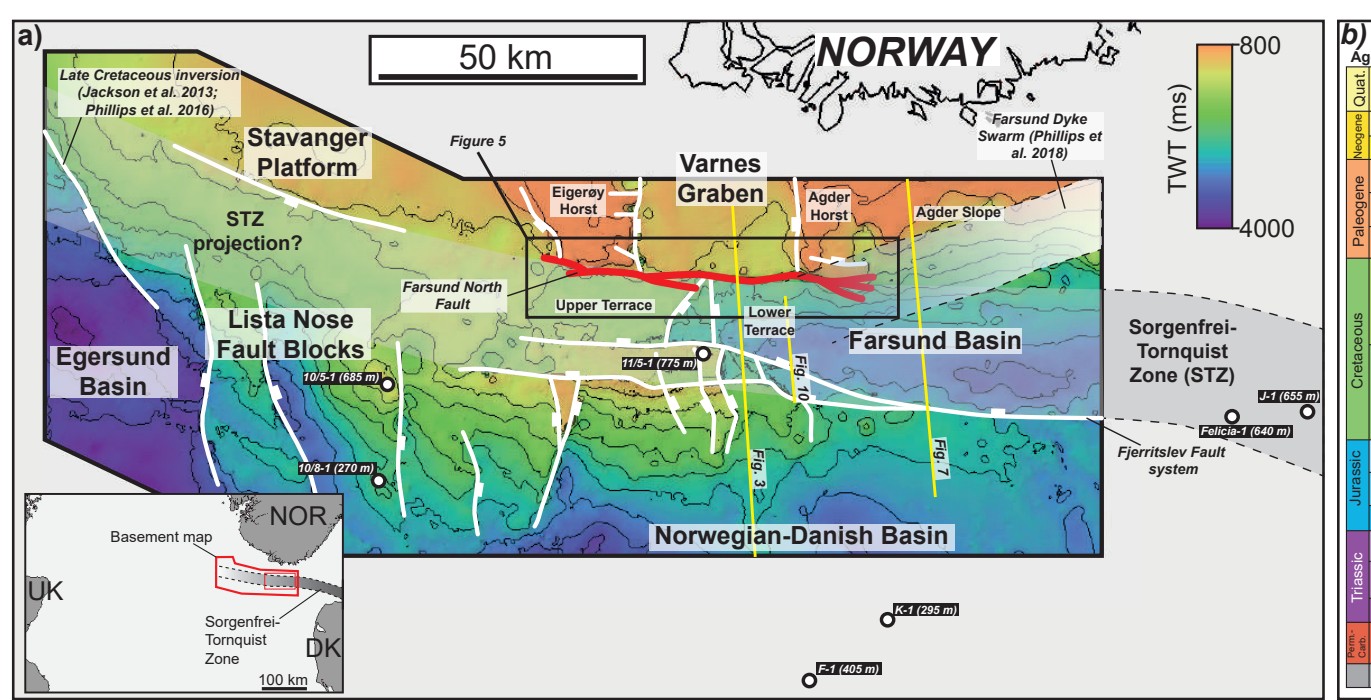

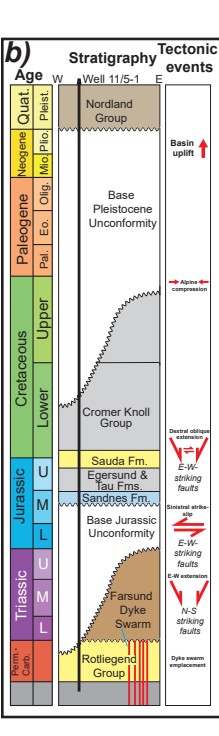

Figure 2



**End Triassic**

Figure 3



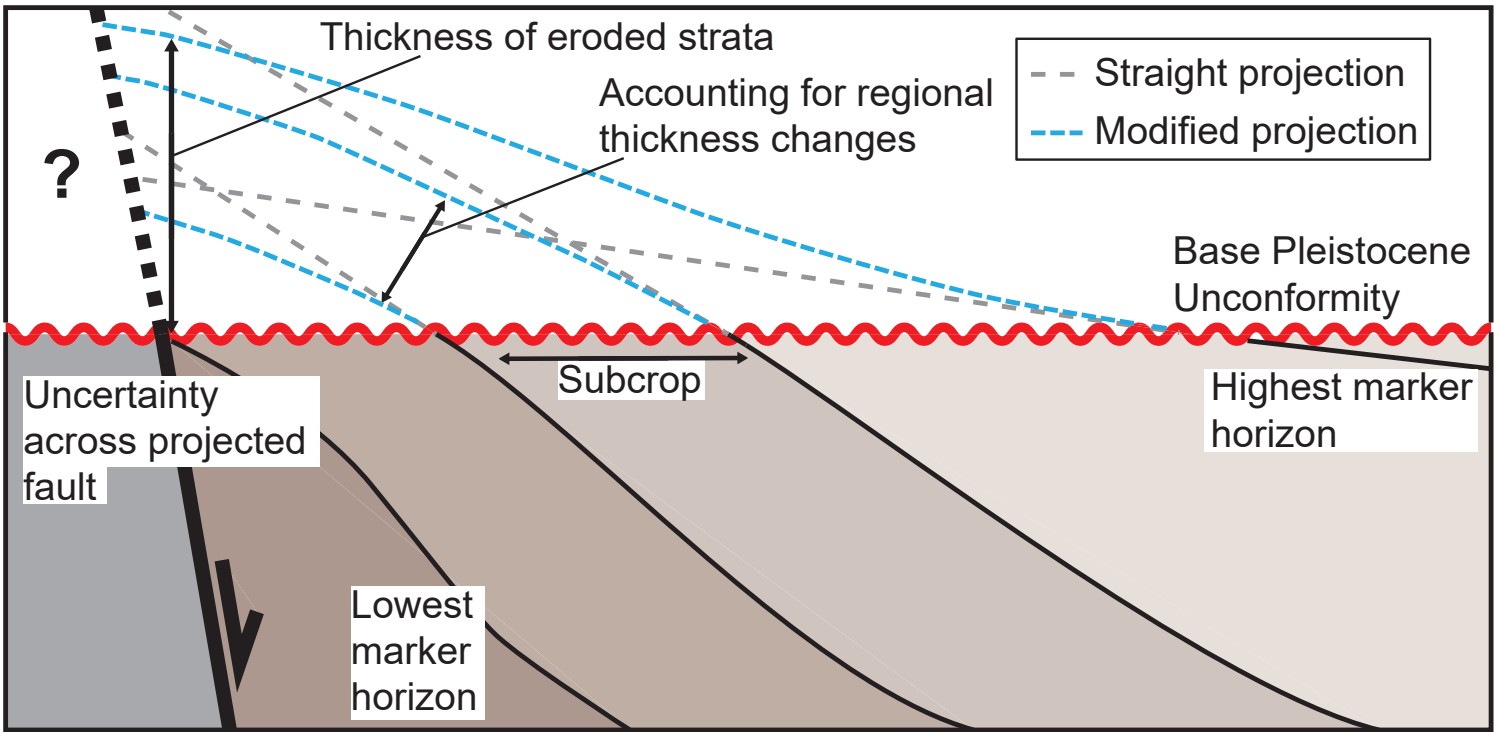

Figure 4



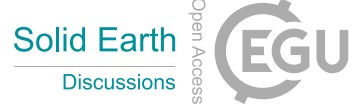



# Figure 5



| Well | Logged section (m burial depth) | Average $E_N$ (m) | $B_E$ (m) | $E_G$ (m) | Japsen and Bidstrup, 1999 | | Japsen et al, 2007 | |
|---|---|---|---|---|---|---|---|---|
| | | | | | $E_G$ | $E_G$ difference | $E_G$ | $E_G$ difference |
| 11/5-1 | 215 - 830 | 605 | 170 | **775** | - | - | - | - |
| 10/8-1 | 1075 - 1180 | 195 | 75 | **270** | 600 | 330 | - | - |
| 10/5-1 | 1090 - 1200 | 570 | 115 | **685** | 500 | 185 | - | - |
| K-1 | 1080 - 1240 | 240 | 55 | **295** | 600 | 305 | - | - |
| F-1 | 1290 - 1440 | 385 | 20 | **405** | 500 | 95 | - | - |
| J-1 | 170 - 740 | 640 | 15 | **655** | 600 | 55 | - | - |
| Felicia-1 | 640 - 830 | 610 | 30 | **640** | 800 | 160 | 900 | 260 |

Figure 6



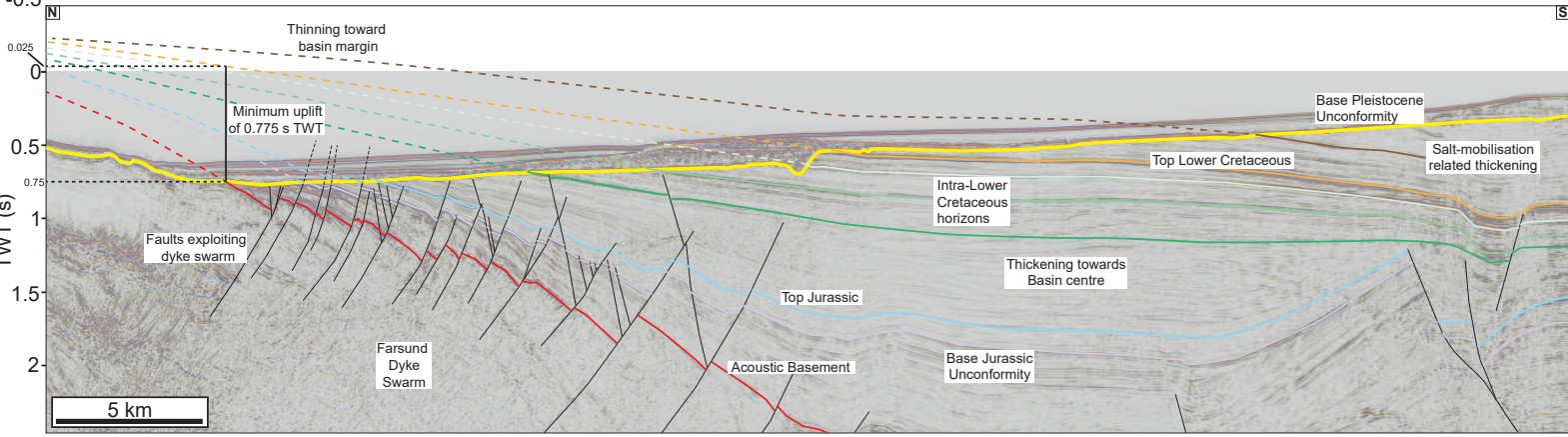

Figure 7



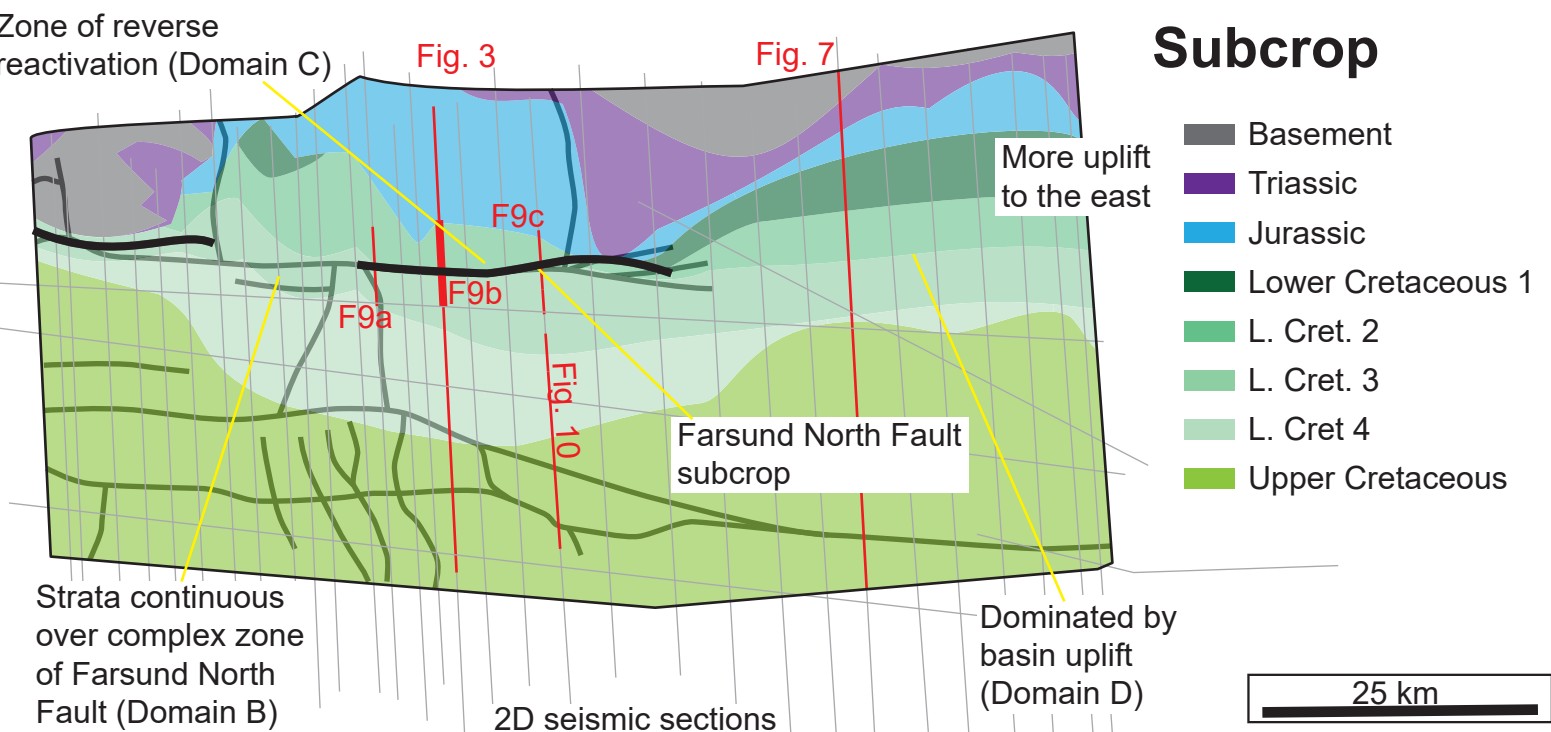

Figure 8

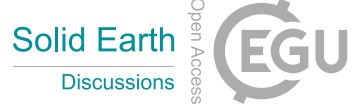

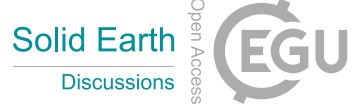

Figure 9



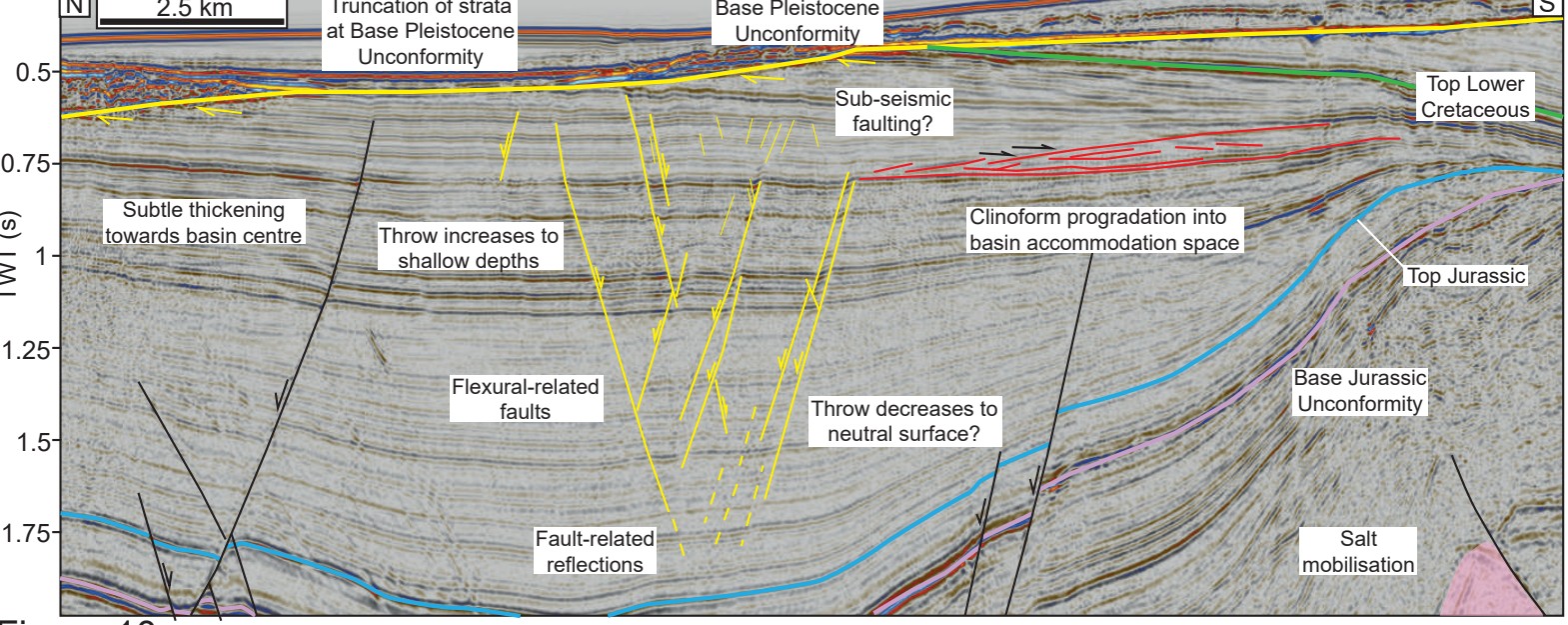

Figure 10







Figure 11

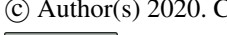

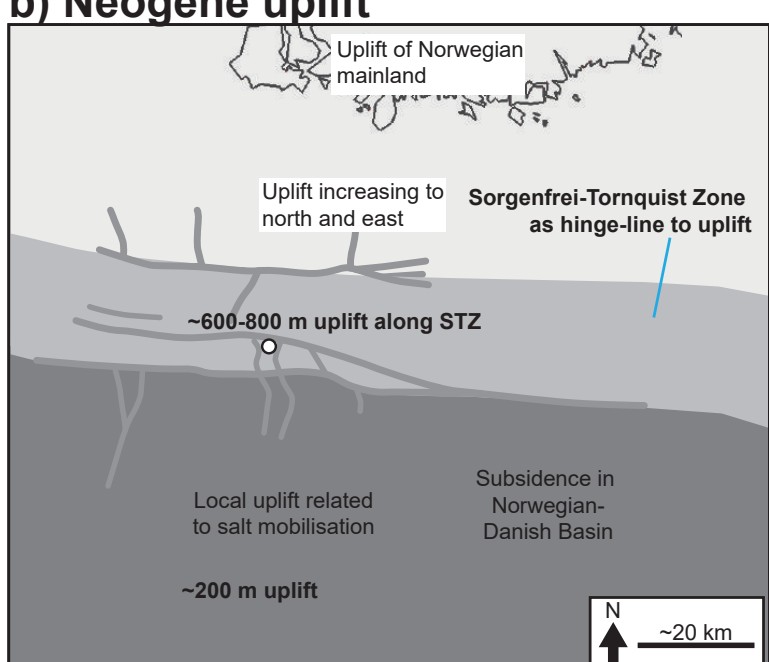

## a) Late Cretaceous inversion

Reverse reactivation of
planar section of
Farsund North Fault

**Compression localised along
the Sorgenfrei-Tornquist Zone**

Basin uplift and
flexural faulting

Inversion localised along
south-dipping fault

Alpine Orogeny
compression

N    ~20 km

## b) Neogene uplift

Uplift of Norwegian
mainland

Uplift increasing to
north and east

**Sorgenfrei-Tornquist Zone
as hinge-line to uplift**

**~600-800 m uplift along STZ**

Local uplift related
to salt mobilisation

Subsidence in
Norwegian-
Danish Basin

**~200 m uplift**

N    ~20 km

Figure 12