# Peer review of "Pre-inversion normal fault geometry controls inversion style and magnitude, Farsund Basin, offshore southern Norway"

_Solid Earth, 2020_

## Referee Comment (RC1) · Pablo Rodríguez Salgado (Referee) · 6 Apr 2020

This manuscript focuses on the style and mechanisms of inversion in the northern margin of the Farsund Basin (south Offshore Norway) throughout Late Cretaceous and Cenozoic compressional events. The authors provide a detailed analysis of the basin bounding Farsund North Fault based on 2D and 3D seismic reflection data. In addition, the authors perform exhumation estimates from stratigraphic and compaction methods by using seismic and well data, respectively. The authors show i) how the pre-inversion structural configuration on the Farsund North Fault controls along-strike variations on the subsequent inversion-related deformation and ii) that the magnitude of uplift in-

creases towards the Sorgenfrei-Tornquist Zone (a major lithosphere structure) which acts as a hingeline separating areas of regional uplift during the Cenozoic in the north from areas of subsidence in the south. The manuscript is well written, and the figures are of excellent quality reflecting all the points described in the text. I consider that this manuscript contributes to improve the general understanding of the mechanisms and styles of basin inversion. More specifically, this manuscript contributes with new data and observations to the knowledge of the Late Cretaceous – Cenozoic basin inversion and exhumation episodes in the Norwegian-Danish continental shelf. For that reason, I recommend publication of the manuscript after addressing few minor revisions.

1) Line 284 – "This normal compaction curve, which assumes continuous burial and hydrostatic stress conditions". Does that mean that the formation pressure of lower Cretaceous interval is hydrostatic in the 7 wells selected for this study? If so, the authors should mention it in the methodology section (subsection 3.4.2 Well-based compaction analysis). Also, have the authors performed any pre-conditioning of the sonic log curve (e.g. removal of anomalous DT readings)? I think that this should also be mentioned in the methodology section.

2) Line 347 – "Domain B is characterised by a complex zone of faulting formed during Early-Middle Jurassic strike-slip faulting (Phillips et al., 2018)". The authors mention earlier in the text (line 107) that the strike-slip fault system developed during the Jurassic takes place along E-W-striking faults and therefore, they are parallel to the Farsund North Fault. Can the authors provide further details about the geometry and orientation of the strike-slip fault zone along the Farsund North Fault? Although some details are given in the line 257 "The proposed strike-slip fault continues towards Domain A to the west, and continues to the southeast, south of Domain C, to the east." I think the manuscript could be improved by indicating the location of the strike-slip faults in the Figure 5a.

3) Line 270 – "We suggest that Domain C represents an Early Cretaceous segment of the Farsund North Fault, which propagated away from a pre-existing segment (Domain

A) during the Early Cretaceous, with Domain B situated between the two segments."
I completely agree. However, I think that segmentation along the Farsund North Fault
is an important point and it could be further developed in the text. Is there any evidence of fault segmentation in the throw-distance profile shown in the Fig. 5b? For
example, displacement patterns in this figure show throw maxima in the central part of
the domains A and C decreasing towards the domain B. Also, a similar throw pattern
is observed between the domains C and D. Could these patterns reflect different kinematically linked segments? Is there any relay ramp observed between the domains A
and C and the domains C and D? In the TWT structure map (Fig 5a) the authors show
the Farsund North Fault as a continuous structure. Can the authors show the fault
segments mentioned in the text (e.g. lines 270 and 349)? Also, can these segments
be shown in the interpreted seismic sections (Figs 5a and 8a)?

4) Line 349 – "The eastern fault segment (Domain C) only initiated in the Early Cretaceous, with Carboniferous-Permian strata being isopachous across the fault (Fig. 3).
The western segment of the Farsund North Fault was also active during Early Cretaceous extension and may have been active earlier during Carboniferous-Permian extension, although we are unable to confirm this due to a lack of preserved strata".
I understand that whereas both segments were active at least since Early Cretaceous times, there is no evidence that the Farsund North Fault was active during
Carboniferous-Permian times. Is there any other evidence of Carboniferous-Permian
activity (i.e. growth of the sequences) recorded by any other E-W-striking faults in the
area? I think this could be an important point to be added to the conclusions.

5) Is there a null-point observed in any of the Lower Cretaceous or older horizons along
the Farsund North Fault? I think it is worth mentioning in the manuscript whether this
observation is made or not.

6) Whereas the authors interpret a Late Cretaceous age for the inversion and exhumation in the Farsund Basin (Line 419) I understand that magmatic underplating is the
main uplift mechanism behind the Neogene exhumation episode (e.g. paragraph starting in Line 425). However, in some parts of the manuscript the authors include both the Late Cretaceous and Neogene episodes within the term basin inversion. Some example are in the line 413: "We suggest the amplitude of the fold at shallow depths is more representative of the structural style forming during Late Cretaceous-Neogene inversion" or in the line 395: "The Farsund Basin experienced at least two phases of inversion during the Late Cretaceous and the Neogene". As the term basin inversion implies that uplift is controlled by reverse reactivation of a pre-existing fault system (Cooper et al., 1989). This excludes any other source of uplift not caused by compressional reactivation of pre-existing faults (Chadwick et al., 1993) I think therefore that the use of the term basin inversion in some parts of the manuscript should be revised.

Other comments:

Line 38: References missing: Gontijo-Pascutti et al., 2010 and Chattopadhyay and Chakra, 2013

Line 99: References missing: Jackson and Lewis, 2013

Line 193: Reference Japsen et al., 2007a or Japsen et al., 2007b? I think it is Japsen et al., 2007a. The same in the figure 6b.

Figure 5a: Which horizon corresponds to the TWT structure map? It is not mentioned in the figure nor in the caption.

References: Cooper, M. A., Williams, G. D., De Graciansky, P. C., Murphy, R. W., Needham, T., De Paor, D., ... & Ziegler, P. A. (1989). Inversion tectonics—a discussion. Geological Society, London, Special Publications, 44(1), 335-347. Chadwick, R. A. (1993). Aspects of basin inversion in southern Britain. Journal of the Geological Society, 150(2), 311-32

---

## Referee Comment (RC2) · Fabian Jähne-Klingberg (Referee) · 12 May 2020

In this publication the authors deal with faults and inverted graben structures along the western part of the Sorgenfrei Tornquist Zone (STZ). A prominent basin in this region is the Lower Cretaceous Farsund Basin. Within the manuscript the authors interpret on the base of a comprehensive seismic mapping study and additional well analysis the structural history of this structure and especially focus in detail on the uplift and exhumation history. The differences of two in structural style different inversion/uplift events were described in detail - the Late Cretaceous and a Neogene event. The estimates of uplift and erosion of this region provide important information for assessing

the importance of the STZ during the Late Cretaceous and Neogene in the context of the Central European Basin System. Furthermore, the authors show that the structures in their study area are segmented and that the structural inversion had different effects on individual sections of the Farsund basin. Possible causes for that are discussed in detail in the manuscript. The subject matter is well presented in the manuscript. Most illustrations contribute to understanding of the manuscript in their current form. The manuscript provides important new insights into the structural development along the STZ. For this reason I recommend publishing the manuscript after moderate revisions have been made.

Detailed comments can be found in the annotated PDF in the supplement. The main points of the review are briefly summarised below:

1.) The authors should explain more clearly their definition of "inversion" and "reactivation". Are all the interpreted parts of the structure which show uplift/erosion are inverted? Or is inversion one structural style of shortening along the whole structure? Is inversion the effect of shortening or as well of other processes? Is the term "inversion" used as umbrella for structural inversion as well for basin inversion? Compressional/transpressional reactivation/shortening is perhaps the better umbrella term. Unfortunately not so handy in the description. Inversion as reverse reactivation of faults, and uplift of grabens & basins is not the only effect of Upper Cretaceous shortening. In some areas only basement flexures and steep reverse-faults can be detected.

2.) Can the uplift, especially the Neogene, be explained by other processes as shortening (e.g. dynamic topography)? And can the Neogene Uplift which is not clearly related to structures or basins really be called inversion?

3.) Not the Alpine-Carpathian orogeny is the reason for Late Cretaceous shortening of structures in the CEBS, but it's the result from Africa-Iberia-Europe convergence. Greater parts of the alps show an extensional setting during the Cretaceous. But the Pyrenees were active during this time.

REF: -Kley, J. & Voigt, T. (2008): Late Cretaceous intraplate thrusting in central Europe: Effect of Africa-Iberia-Europe convergence, not Alpine collision. Geology, 36, 11: pp. 839-842. DOI:10.1130/g24930a.1

4.) Please avoid "unclear relations", generalisations like: - Late Cretaceous inversion (Late Cretaceous shortening has not only produced inversion structures.) - Neogene shortening (the mechanism behind Neogene uplift is still under discussion) - Alpine compressional stresses - Alpine inversion - Late Cretaceous-Neogene inversion (The link with hyphen is misleading, since there were long pauses between events.)

5.) What is with in literature described indications of Paleogene uplift of this region?

REF e.g.: -Clausen, O.R., Nielsen, O.B., Huuse, M. & Michelsen, O. (2000): Geological indications for Palaeogene uplift in the eastern North Sea Basin. Global and Planetary Change, 24: pp. 175-187.

-Japsen, P., Green, P.F., Nielsen, L.H., Rasmussen, E.S., Bidstrup, T., 2007a. Mesozoic–Cenozoic exhumation events in the eastern North Sea Basin: a multi-‐disciplinary study based on palaeothermal, palaeoburial, stratigraphic and seismic data. Basin Res 19, 451-490.

Please discuss more the definition and proof of the uplift events presented here. Are others to be excluded or just not to be recognized or verified by the applied analytics?

6.) a map of data coverage in relation to the interpreted structures would be useful

7.) some questions of understanding to the data & methods chapter (see comments in the annotated PDF in the supplement). If possible, please make additional insertions in the text for better understanding.

8.) To "6.3 Localisation of inversion along pre-existing structures" line 448: Huyghe & Mugnier (1994, 1995) point to the relationship between rifting, the time elapsed thereafter and the potential for reactivation/inversion of the structure. So maybe the fault-reactivation and structural inversion with anticlinal folding is a consequence that

compression in a properly aligned vector meets with the Farsund Basin a young"fresh" graben.

REF: -Huyghe, P. & Mugnier, J.L. (1995): A comparison of inverted basins of the Southern North Sea and inverted structures of the external Alps. (In: Buchanan, J.G. & Buchanan, P.G. (Eds.): Basin Inversion). Vol. 88: pp. 339-353; London (Geological Society Special Publication).

-Huyghe, P. & Mugnier, J.L. (1994): Intra-plate stresses and basin inversion: A case from the Southern North Sea. (In: Roure, F. (Ed.): Peri-Tethyan platforms). pp. 211-226; - (Éditions Technip).

-Huyghe, P. & Mugnier, J.L. (1992): Short-cut geometry during structural inversions; competition between faulting and reactivation. Bulletin de la Société Géologique de France, 163, 6: pp. 691-700.

8.) "488-490: We suggest that the likelihood of a structure to be reactivated and undergo inversion is not solely related to the size and 'weakness' of the structure; the relative complexity of the structure also plays an important role."

Your presented seismic profiles show only the top 6 sec. twt of the strat. column. What information do the authors have about the geometry and complexity of the fault with the depth. A complex fault pattern in the most uppern strat. column does not have to mean that the fault in the deeper section must have a complex geometry.

9.) Can statements about the amount of shortening during the Late Cretaceous and during "the Neogene" (if this event is related to shortening) be made?

10.) Various comments on illustrations (Please see the annotated PDF in the supplement). - Figure 3 and 11 in particular should be adapted.

Please also note the supplement to this comment:
https://www.solid-earth-discuss.net/se-2020-27/se-2020-27-RC2-supplement.pdf

[Figure]

**Supplement:**

[revised manuscript text omitted]

Figure 1

[Figure]

[Figure]

[Figure]

Figure 2

[Figure]

[Figure]

Figure 3

[Figure]

[Figure]

Figure 4

[Figure]

[Figure]

Figure 5

[Figure]

**a)**

[Figure]

**b)**

| Well | Logged section (m burial depth) | Average $E_N$ (m) | $B_E$ (m) | $E_G$ (m) | Japsen and Bidstrup, 1999 | | Japsen et al, 2007 | |
|---|---|---|---|---|---|---|---|---|
| | | | | | $E_G$ | $E_G$ difference | $E_G$ | $E_G$ difference |
| 11/5-1 | 215 - 830 | 605 | 170 | **775** | - | - | - | - |
| 10/8-1 | 1075 - 1180 | 195 | 75 | **270** | 600 | 330 | - | - |
| 10/5-1 | 1090 - 1200 | 570 | 115 | **685** | 500 | 185 | - | - |
| K-1 | 1080 - 1240 | 240 | 55 | **295** | 600 | 305 | - | - |
| F-1 | 1290 - 1440 | 385 | 20 | **405** | 500 | 95 | - | - |
| J-1 | 170 - 740 | 640 | 15 | **655** | 600 | 55 | - | - |
| Felicia-1 | 640 - 830 | 610 | 30 | **640** | 800 | 160 | 900 | 260 |

Figure 6

[Figure]

[Figure]

Figure 7

[Figure]

[Figure]

Figure 8

[Figure]

[Figure]

Figure 9

[Figure]

[Figure]

Figure 10

[Figure]

[Figure]

**Figure 11**

[Figure]

[Figure]

**a) Late Cretaceous inversion**

Reverse reactivation of
planar section of
Farsund North Fault

**Compression localised along
the Sorgenfrei-Tornquist Zone**

Basin uplift and
flexural faulting

Inversion localised along
south-dipping fault

Alpine Orogeny
compression

N     ~20 km

**b) Neogene uplift**

Uplift of Norwegian
mainland

Uplift increasing to
north and east

**Sorgenfrei-Tornquist Zone
as hinge-line to uplift**

**~600-800 m uplift along STZ**

Local uplift related
to salt mobilisation

Subsidence in
Norwegian-
Danish Basin

**~200 m uplift**

N     ~20 km

Figure 12

---

## Author Comment (AC1) · 9 Jun 2020

Response to Review comments 1 – Pablo Rodriguez-Salgado

We thank the reviewer for their overall positive comment on the manuscript

"I consider that this manuscript contributes to improve the general understanding of the mechanisms and styles of basin inversion. More specifically, this manuscript contributes with new data and observations to the knowledge of the Late Cretaceous – Cenozoic basin inversion and exhumation episodes in the Norwegian-Danish continental shelf. For that reason, I recommend publication of the manuscript after addressing

few minor revisions."

We have responded to the individual comments and points raised by the reviewer below and included the changes in the revised manuscript, as shown in the attached track changes document. Our responses are shown in italics with line numbers corresponding to changes in the track changes document with full markup shown.

Reviewer comments and responses:

1) Is the formation pressure of lower Cretaceous interval is hydrostatic in the 7 wells selected for this study? Have the authors performed any pre-conditioning of the sonic log curve (e.g. removal of anomalous DT readings)?

Response

No overpressure was recorded within the Lower Cretaceous interval across the wells used in this study, hence we assumed that the Lower Cretaceous interval formation pressure was hydrostatic. This is consistent with regional studies which also show no signs of overpressure in this interval (e.g. Japsen et al., 1998). Additional text, explaining the hydrostatic nature of the wells has been added to the revised manuscript (Line 245-249) The sonic log curve was pre-conditioned prior to being incorporated into our analyses (Line 268-269) in order to remove any anomalous values. We have now clarified the text on lines 255-258 to better illustrate the pre-conditioning of the log prior to analysis.

2) Can the authors provide further details about the geometry and orientation of the strike-slip fault zone along the Farsund North Fault? Although some details are given in the line 257 "The proposed strike-slip fault continues towards Domain A to the west, and continues to the southeast, south of Domain C, to the east." I think the manuscript could be improved by indicating the location of the strike-slip faults in the Figure 5a.

Response

Due to erosion at the base Jurassic unconformity, which removes related growth strata, the presence of the strike slip faults can only be determined by the offsets between older (i.e. Permo-Triassic), N-S- striking faults. We have revised manuscript to make this clearer (Line 128-130).

We can confirm the location of an older (i.e. pre-Cretaceous) strike-slip fault between offset N-S-striking faults, i.e. the fault partitioning the Farsund Basin and the fault along the western margin of the Varnes Graben. However, we can also show that no pre-cursor (i.e. Early-Middle Jurassic), strike-slip faulting occurred along the eastern segment of the Farsund North Fault, which initiated during the Early Cretaceous; we therefore suggest that the strike-slip system continued eastwards with a NW trend. A similar relationship occurs along the southern margin of the Farsund Basin, with the strike-slip system, the position and offset of which is constrained by the offset N-S-striking faults, interpreted to continue north of an Early Cretaceous segment of the Fjerritslev South Fault with a NW-SE strike (Phillips et al., 2018). Additional information has been included in the manuscript (Lines 298-313, 320-330) to better illustrate our constraints on the overall geometry of the strike-slip fault system. We also link our observations made here from the northern basin margin to complementary observations from the southern basin margin (see Phillips et al., 2018).

We have added a proposed continuation of the strike-slip system to Figure 5a to make the relationship between it and the eastern segment of the Farsund North Fault (Domain C) clearer.

3) Line 270 – "We suggest that Domain C represents an Early Cretaceous segment of the Farsund North Fault, which propagated away from a pre-existing segment (Domain A) during the Early Cretaceous, with Domain B situated between the two segments." I completely agree. However, I think that segmentation along the Farsund North Fault is an important point and it could be further developed in the text. Is there any evidence of fault segmentation in the throw-distance profile shown in the Fig. 5b? For example, displacement patterns in this figure show throw maxima in the central part of the domains A and C decreasing towards the domain B. Also, a similar throw pattern is observed between the domains C and D. Could these patterns reflect different kinematically linked segments? Is there any relay ramp observed between the domains A and C and the domains C and D? In the TWT structure map (Fig 5a) the authors show the Farsund North Fault as a continuous structure. Can the authors show the fault segments mentioned in the text (e.g. lines 270 and 349)? Also, can these segments be shown in the interpreted seismic sections (Figs 5a and 8a)?

Response

We agree that segmentation along the Farsund North Fault, as represented by the present distribution of throw, is an important point that was underexplored in the initial manuscript. In the revised manuscript we discuss in greater detail the geometry and distribution of throw along this fault 1 (Lines 299-301; 344-348).

We note throw maxima in the centre of domains A and C, and suggest that these likely reflect segmentation of the initial fault, particularly as the eastern fault segment did not reactivate a pre-existing strike slip fault (see response to point 2). Relay ramp segmentation is likely present between Domain B and Domain C. We suggest that the segment in Domain C propagated away from Domain B (where we know a strike-slip precursor fault was present, see point 2) during the Early Cretaceous, rather than representing a segment that subsequently linked with the one in Domain B.

Domain D largely covers the Agder slope, east of the Farsund North Fault, although the eastern termination of the eastern fault segment (i.e. the segment that characterises Domain C) extends into the western end of Domain D (Line 351).

4) I understand that whereas both (fault) segments were active at least since Early Cretaceous times, there is no evidence that the Farsund North Fault was active during Carboniferous-Permian times. Is there any other evidence of Carboniferous-Permian activity (i.e. growth of the sequences) recorded by any other E-W-striking faults in the area? I think this could be an important point to be added to the conclusions.

Response

Extensional faults defining the Farsund Basin are largely not present during pre-Zechstein extension (See Figure 3), in contrast to elsewhere along the Sorgenfrei-Tornquist Zone, where faults were active during Carboniferous-Permian extension. We can also confirm that the eastern segment of the Farsund North Fault was Early Cretaceous or younger in age.

We note that the western segment of the Farsund North Fault was likely active during Early-Middle Jurassic strike-slip activity, with the Farsund North Fault showing a similar geometric and kinematic relationship to that of the Fjerritslev Fault System along the southern basin margin and propagating away from the strike-slip structure (see response to point 2). However, due to erosion at the BJU across the Upper Terrace of the Farsund Basin, we are unable to determine whether this area and associated faults experienced earlier, Carboniferous-Permian activity. We note that Carboniferous-Permian extension has been documented elsewhere along the Tornquist Zone (e.g. Erlstrom et al., 1997), and is also documented to the west in the Egersund Basin (Jackson and Lewis, 2013) (Line 117).

We have revised the manuscript to better outline the evolutionary histories of the various fault segments and to also better convey the uncertainty regarding Carboniferous-Permian activity along the Farsund North Fault, particularly along the western segment (Line 322-325, 442-445).

5) Is there a null-point observed in any of the Lower Cretaceous or older horizons along the Farsund North Fault? I think it is worth mentioning in the manuscript whether this observation is made or not.

Response

We do not identify a null point at any point along the Farsund North Fault as the magnitude of the initial extensional offset along the fault was much greater than the subsequent reverse offset. Therefore, the fault displays net-extensional offset at all depths. We have included this important point in the revised manuscript (Line 417-419)

6) Whereas the authors interpret a Late Cretaceous age for the inversion and exhumation in the Farsund Basin (Line 419) I understand that magmatic underplating is the main uplift mechanism behind the Neogene exhumation episode (e.g. paragraph starting in Line 425). However, in some parts of the manuscript the authors include both the Late Cretaceous and Neogene episodes within the term basin inversion. As the term basin inversion implies that uplift is controlled by reverse reactivation of a pre-existing fault system (Cooper et al., 1989). This excludes any other source of uplift not caused by compressional reactivation of pre-existing faults (Chadwick et al., 1993) I think therefore that the use of the term basin inversion in some parts of the manuscript should be revised.

Response

We agree with the points raised by the reviewer here that the term "basin inversion" should not be used to describe the Neogene uplift event, and that this event should be clearly distinguished from the earlier, Cretaceous event. Furthermore, as we are unable to distinguish between individual uplift pulses, including those that may have occurred earlier in the Paleogene, we now refer to 'Paleogene-Neogene uplift' rather than 'Neogene inversion' to encompass all post Cretaceous uplift (e.g. lines 102, 138, 495). We also agree with the reviewer regarding our mode general use of the term "basin inversion".

Following additional comments from reviewer 2, we have modified our usage of this term. We now refer to Late Cretaceous "compression" or "shortening", which we argue were expressed via a number of different mechanisms across the Farsund Basin, including basin inversion (which was explicitly associated reverse reactivation of basin-bounding normal faults). This has been modified in the revised manuscript (e.g. Lines 30-42).

Additional minor and textual changes have been completed throughout the manuscript.

Please also note the supplement to this comment:
https://se.copernicus.org/preprints/se-2020-27/se-2020-27-AC1-supplement.pdf

―――――――――――――――――――

[Figure]

[Figure]

**Fig. 1.** Revised Figure 5

**Supplement:**

[revised manuscript text omitted]

---

## Author Comment (AC2) · 9 Jun 2020

Review 2 – Fabian Jahne-Klingberg

We thank the reviewer for their detailed and thorough review of the manuscript and their positive comments:

"The subject matter is well presented in the manuscript. Most illustrations contribute to understanding of the manuscript in their current form. The manuscript provides important new insights into the structural development along the STZ. For this reason I recommend publishing the manuscript after moderate revisions have been made."

[Figure]

We have responded to the individual comments made by the reviewer below and have made changes accordingly in the track changes document. Our responses are shown in italics with line number corresponding to the full markup document. Revised figures are also attached to this document. We believe that these comments and our associated changes have greatly improved the manuscript.

Many thanks, Thomas Phillips

Reviewer comments and responses:

1. The authors should explain more clearly their definition of "inversion" and "reactivation". Are all the interpreted parts of the structure which show uplift/erosion are inverted? Or is inversion one structural style of shortening along the whole structure? Is inversion the effect of shortening or as well of other processes? Is the term "inversion" used as umbrella for structural inversion as well for basin inversion? Compressional/transpressional reactivation/shortening is perhaps the better umbrella term.

Response - See also response to point 6 of reviewer 1. We have amended the terms that we use throughout the manuscript. We agree that 'inversion' represents just one mechanism that accommodates shortening along the northern basin margin. We have followed the definition from Williams and Turner (1989) with regards to inversion being the process by which previously extensional structures experience uplift and compression (Line 30-32). We have modified the manuscript, particularly in the introduction to state the different mechanisms by which compression and shortening can be accommodated, including the reverse reactivation of faults, and the inversion of previously extensional basins (Line 32-34). Specific changes can be found in response to the comments from the annotated pdf.

2. Can the uplift, especially the Neogene, be explained by other processes as shortening (e.g. dynamic topography)? And can the Neogene Uplift which is not clearly related to structures or basins really be called inversion?

Response - We agree that the Paleogene-Neogene uplift is not necessarily related to inversion and shortening, and have since modified this in the revised manuscript in response to other comments (i.e. points 1, 4). As we are unable to distinguish between individual uplift events throughout the Cenozoic we now refer to these uplift phases collectively as Paleogene-Neogene uplift. We outline some proposed causes of the Paleogene and Neogene uplift events, including upper mantle motions and dynamic topography (Line 150) and also plate tectonic forces associated with the opening of the North Atlantic (Line 150-151). We note that whilst Late Cretaceous compression is amplified along the STZ, the STZ represents a relative hingeline during Paleogene-Neogene uplift, separating areas of relatively high and low uplift.

3. Not the Alpine-Carpathian orogeny is the reason for Late Cretaceous shortening of structures in the CEBS, but it's the result from Africa-Iberia-Europe convergence. Greater parts of the alps show an extensional setting during the Cretaceous. But the Pyrenees were active during this time.

Response - We thank the reviewer for raising this point that it is not the Alpine-Carpathian orogeny that is responsible for the Late Cretaceous compression, rather the convergence of Africa, Iberia and Europe, as outlined by Kley and Voigt (2008). As such, we now relate Late Cretaceous compression to Africa-Iberia-Europe convergence rather than the Alpine Orogeny throughout the manuscript (e.g. Line 83-84, 140, 502, 545).

4. Please avoid "unclear relations", generalisations like: - Late Cretaceous inversion (Late Cretaceous shortening has not only produced inversion structures.) – Neogene shortening (the mechanism behind Neogene uplift is still under discussion) – Alpine compressional stresses - Alpine inversion - Late Cretaceous-Neogene inversion (The link with hyphen is misleading, since there were long pauses between events.)

Response - We agree with the reviewer. Late Cretaceous shortening is expressed differently along the STZ via a number of different mechanisms. Within the Farsund Basin

in particular, we show that this shortening is expressed via the reverse reactivation of normal faults (i.e. basin inversion), long wavelength folding of the basin fill and potentially regional uplift. After the points raised and changes made in response to point 1 of the reviewer, we now use Late Cretaceous shortening/compression and Paleogene-Neogene uplift (Line 138) to refer to the two main events in this study. Within these overarching events, we are more explicit and refer directly to the different mechanisms, i.e. reverse fault reactivation etc., occurring along different parts of the northern basin margin. As a result of these changes and an overall restructuring and clarification of the terms used in the manuscript, we have amended these "unclear relations" and "generalisations".

5. What is with in literature described indications of Paleogene uplift of this region?

Response Because Uppermost Upper Cretaceous-Neogene strata are absent across the Farsund Basin, we are unable to distinguish individual Neogene and Paleogene uplift events, which we now refer to collectively as 'Paleogene-Neogene uplift,' (see responses above). We have clarified this in the revised manuscript (e.g. Line 154). We have also included additional information relating to the potential causes of these uplift events (Line 146-152). Because strata and unconformities related to the Paleogene-Neogene uplift event(s) are absent, we are unable to speculate as to the exact cause and timing of these uplift events, which we argue lie outside of the scope of this study.

6. a map of data coverage in relation to the interpreted structures would be useful

Response A map of the seismic data referred to in this study is shown in Figure 8. We have also included the locations of the seismic sections and 3D seismic volume for the main study area on the revised Figure 2.

7. some questions of understanding to the data & methods chapter (see comments in the annotated PDF in the supplement). If possible, please make additional insertions in the text for better understanding.

Response Specific points in response to comments raised in the annotated pdf are shown below.

8. line 448: Huyghe & Mugnier (1994, 1995) point to the relationship between rifting, the time elapsed thereafter and the potential for reactivation/inversion of the structure. So maybe the fault-reactivation and structural inversion with anticlinal folding is a consequence that compression in a properly aligned vector meets with the Farsund Basin a young "fresh" graben.

Response We thank the reviewer for raising this interesting point regarding the young age of the Farsund Basin compared to other rift systems along the STZ. We agree that the short turnaround between extension and compression may increase the likelihood of reactivation within the Farsund Basin. Specifically, we note that the eastern segment of the Farsund North Fault, which formed only during Early Cretaceous rifting, may be more prone to reactivation than older structures. We have incorporated some additional text discussing this idea, as proposed in Huyghe and Mugnier (1995), to lines 558-564. We also incorporate some additional information regarding the easterly rather than south-easterly trend of the Farsund Basin.

9. "488-490: We suggest that the likelihood of a structure to be reactivated and undergo inversion is not solely related to the size and 'weakness' of the structure; the relative complexity of the structure also plays an important role." Your presented seismic profiles show only the top 6 sec. twt of the strat. column. What information do the authors have about the geometry and complexity of the fault with the depth. A complex fault pattern in the most uppern strat. column does not have to mean that the fault in the deeper section must have a complex geometry.

Response We agree that a complex fault pattern at shallow depths does not indicate a complex fault at depth. We are typically unable to identify any complex fault geometries at depth. We interpret that the Farsund North Fault is defined by a single planar fault geometry at depth. We have amended the sentence in question to highlight that the

along-strike complexity appears to represent the key aspect as to whether the structure was inverted (Line 336-338, 605)

10. Can statements about the amount of shortening during the Late Cretaceous and during "the Neogene" (if this event is related to shortening) be made?

Response We are unable to comment on the absolute values of shortening that occurred during the Late Cretaceous. However, we do note that the amount of shortening was relatively mild and of a similar magnitude to that observed further west along the Stavanger Fault System, in the Egersund Basin (Line 517-518). Well-based compaction analyses highlight the bulk amount of uplift that occurred in the immediate vicinity of the well, however, we are unable to distinguish between the distinct uplift events and phases that occurred. In response to earlier comments raised by the reviewer (see points 1 and 4) we have now changed the terminology used such that we no longer refer to "Neogene shortening" and instead refer to "Paleogene-Neogene uplift".

11. Various comments on illustrations (Please see the annotated PDF in the supplement). - Figure 3 and 11 in particular should be adapted.

Response See specific responses below for each figure in response to the comments in the annotated pdf.

Responses to annotated pdf comments – from supplement Minor textual changes and grammatical changes addressed in the pdf have been corrected in the revised manuscript and are shown in the attached track changes document, along with changes already made in response to the preceding points. We here list the more detailed changes made to the manuscript

12. Line 18 – "is it actually shortening or tilting or differential subsidence by other processes (e.g. dynamic topography?) Response We now refer to Paleogene-Neogene uplift as opposed to shortening. We are unable to comment directly on the cause of

the uplift, although we do list some previously proposed mechanisms in Section 2.2.

13. Line 21 – Reactivation is perhaps the better umbrella term (rather than inversion), Inversion of faults, grabens, basins is not only the effect of upper cretaceous compression. In some areas only basement flexures and steep reverse faults along older permo-carboniferous pre-cursors can be seen (SE-Germany, NE-Germany). From this I would define structural inversion itself as a deformation-type-mechanism.

Response We have modified this sentence so that it now reads "how compressional stresses may be accommodated by different mechanisms within structurally complex settings". We acknowledge that these compressional stresses and related basin shortening may be accommodated by different mechanisms. This is emphasised further in the revised Introduction (Line 30-34) and this terminology is applied throughout the manuscript

14. Line 32 – within the southe Permian Basin, structural inversion especially in the top of the Zechstein salt detachment shows often long-wavelength folding of the whole pre-inversion structure "uberpresste Graben" (there exist no good translation, overpressed graben). Therefore some of the structures look a bit like positive flower structures (e.g. Kockel 2003). An effect of shortening of the post-Zechstein along a well-developed detachment.

Response We agree with the reviewer that this represents an important potential mechanism of inversion. We have incorporated this mechanism and associated reference into the mechanism relating to the thin-skinned folding of strata above a detachment (Line 36).

15. Line 35-39 – most of the rift-structures or faults of the pre-Zechstein within the CEBS show steep dipping faults – sometimes near sub-vertical

Response We agree that the majority of the faults in this area are steeply-dipping to sub-vertical. However, in this instance we are referring to pre-existing structures in

general, and under which circumstances they may reactivate. We make no specific reference to the study area at this point in the manuscript.

16. Line 71, 120 –Comment regarding the (potentially earlier) onset of Neogene uplift

We agree with the reviewer that uplift did not solely occur in the Neogene, with earlier events occurring throughout the Paleogene. Following changes made in response to earlier comments we no longer refer to Neogene uplift and instead refer to Paleogene-Neogene uplift events (see responses to Points 2,4 and 5). We have modified the text in these areas to take this into account and added additional references where appropriate (Line 146-154).

17. Line 118 – From Jackson et al., 2013 – the inversion started in the latest Turonian and ceased in the Maastrichtian. More or less the same story as in the whole CEBS with main inversion from Santonian to Campanian. – Check the timing of the event from Jackson et al.

Response We thank the reviewer for pointing this out. We have modified the date of inversion to that referred to in Jackson et al., (2013) (latest Turonian to Early Maastrichtian) (Line 143).

18. Line 125 – Map of the data coverage in relation to the interpreted fault segments would be useful

Response We agree with this point raised by the reviewer and have since added the locations of the 2D seismic sections and 3D seismic volume used in this study to Figure 2, along with figure 8. This figure has also now been referenced accordingly in lines 161 and 165

19. Line 151 – This only minimises errors from the geometrical distortions of dipping structures in the time domain but not the error originating from the general increase in interval-velocities in most lithologies with depth. Therefore only the throw of faults in the same twt interval with more of less the same lithology would be comparable in the

time domain.

Response We agree with the reviewer that measuring throw as opposed to heave does not account for velocity distortions with depth. However, we primarily examine along-strike changes in fault throw, where the lithologies in the hangingwall and footwall remain relatively constant along-strike. Furthermore, we do not focus on the absolute values of throw measured along the faults, rather the overall shape and distribution of throw along the faults. Therefore the changes in throw, which are less influenced by velocity distortions are key to our analysis, and underpin our related conclusions. We have added some text in the revised manuscript to address this point (Line 188-194)

20. Line 162 – Was decompaction taken into account in the course of calculating uplift?

Response Decompaction was accounted for in the well-based calculations of uplift. However, for the seismic-stratigraphic projections, we used a purely geometric approach and did not account for decompaction. The seismic-stratigraphic based approach is aimed at examining the spatial distribution of uplift across the basin rather than absolute values at specific points, as is the case for the well-based calculations. Although decompaction of the strata would change the magnitude of the uplift values calculated here, it would not change the overall spatial pattern. Using a seismic-stratigraphic based approach, we highlight that uplift increases across the basin to the north and to the east; we make no quantitative statements regarding the magnitude of uplift. Furthermore, decompaction would require a depth conversion of the data, which would incorporate more errors into our analyses (Lines 218-224).

21. Line 175 – Was the effect of waterload considered wihtin the calculation? Along the Farsund Basin water depth reaches up to 500 m.b.s.l.

Response The method used in this study only considers the loss of porosity resulting from the thickness of the overlying rock-column, which represents the major factor controlling porosity-loss due to mechanical compaction. Although the waterload will have an effect on the compaction of strata, we do not believe it to be a major factor

(Line 247-248). Furthermore, our approach is consistent with previous studies of well-based uplift in this region (e.g. Japsen et al., 2007), which allows us to directly compare our estimates of uplift to these studies.

22. Line 189 – Mudstone is not just mudstone, there is a great variety of them. The decrease in porosity with depth can also show subtle correlations to, for example, palae-ofacies, hydrostatic pressure conditions (Paleo/Recent), variance in mineral composition. In figure 6, a comparison between wells inside and outside the basin is shown. However the basin and the respective graben parts show strong activity especially during the lower cretaceous. How can other influences on the porosity distribution be excluded? A discussion or explanatory explanations would be nice.

Response This is an interesting point raised by the reviewer. We agree that there is a large variability within mudstones and that other factors may play a role in the compaction of these strata with depth. The regional curve of Hansen et al, (1996) represents the average porosity-depth trend for Cretaceous-Tertiary shales across the Norwegian shelf, and as such likely smooths out these more local features. We have added some additional text to the methods section highlighting how additional factors such as "mineralogy, paleogeographic setting, and burial rate" may affect local porosity-depth trends, but that these small-wavelength variations are largely smoothed out by the average trend of Hansen et al., (1996) (Line 255-258). We further address this point in the results section (Lines 363-366 in response to the reviewer comment on Line 286), where we suggest that the scatter displayed by individual wells may relate to "minor lithological variations, possibly relating to subtle differences in palaeoenvironment and lithological/mineralogical changes between individual wells". The key point is, however, that overcompaction is present regionally in all wells.

23. Line 295, 313 – These estimates (seismic-stratigraphic projections or truncated strata) do not take into account compaction effects and is based only on the actual situation. Regardless of the fact that the analysis was performed in the time domain, the influence of sediment column compaction on the throw should be discussed.

Response We agree that the absolute amplitude of the fold would likely increase following decompaction. Some text has been added to the revised manuscript discussing the effect of compaction on our uplift values in the methods section (Line 222-224). Taking compaction into account, we would expect fold amplitude to decrease with depth due to increased compaction, the opposite to what we observe here, suggesting that decompaction has a limited effect and would only accentuate our current observations. The key point in this section is that fold amplitude changes, both with depth and along-strike, will largely be unaffected by decompaction. We have added text to this effect to the revised manuscript (Line 404-406)

24. Figure 5c – How do you define the base-level for calculation of fold-amplitude in each horizon? Do you see similar changes in the wave-length of the folding in to comparison to the other horizons? (the lateral effect of folding per horizon).

Response Fold amplitude was measured between the fold crest and a local structural datum for each stratigraphic horizon. This local datum was taken as a projection of each stratigraphic horizon from an area unaffected by the near-fault folding. This has been made clearer in the revised manuscript (Line 200-202).

25. Line 351 – Provide references that refer to potential Carboniferous-Permian extension in the area

Response Additional references have been added to this section, detailing Carboniferous-Permian extension occurring elsewhere along the Sorgenfrei-Tornquist Zone and also to the west in the Egersund Basin (Lines 442-445).

26. Line 357 – What kind of reactivation? Is it safe to assume that the entire fault plane has been reactivated?

Response We have changed the phrasing of this section in the revised manuscript to state that the fault underwent "preferential reverse reactivation during Late Cretaceous compression" (Line 453). We assume in this instance that the fault plane was reactivated across all depths, although we are unable to determine whether this was the case, particularly across deeper structural levels.

27. Line 393 – "areas experienced less inversion and therefore preserve the initial monoclinal fold geometry" – What is meant in this context?

Response The fault core and surrounding wall rocks have experienced more deformation and are possibly weaker than areas near the fault tips. As a result, weaker areas near the fault centre would be easier to invert when subject to compression. Thus, assuming the whole length of the fault was subject to the same compressional stress, the weak fault centre would reactivate more readily, and undergo reverse reactivation/slip and related folding of hangingwall strata (see Jackson et al., 2013). We have since altered the wording of this sentence to make this point clearer and relate to the relevant references (Line 489-492).

28. Line 395 – Please define your understanding of inversion/reactivation: Structural inversion related to faults, basin inversion, uplift/exhumation of basins, reactivation.

Response See response to Points 1 and 4 above.

29. Line 395-396 – Can the Neogene uplift/exhumation really be explained by structural inversion? E.g. Kley (2018) Response See response to Points 1 and 4 above. We have added a reference to Kley, (2018) at this point, referring to the Paleogene uplift.

30. Line 402 – the mechanism behind Paleogene uplift/exhumation are under discussion. Refer to Kley et al., 2018

Response This section refers to the compressional phase associated with Africa-Iberia-Europe convergence rather than the later Paleogene-Neogene uplift events. We have discussed the potential mechanisms behind Paleogene-Neogene uplift earlier in the manuscript (section 2.2).

31. Line 431-436 – See previous comments about Paleogene uplift and discuss further (e.g. line 395-396). Prominent refs include Kley et al., 2018

We have incorporated more information relating to Paleogene uplift at the start of section 6.2 and 2.2. However, as we are unable to distinguish between individual uplift events we do not go into detail as to the causes of each uplift event. In the sentence in question (now 536-538) we refer to the specific uplift of the South Scandes and South Swedish domes, which we interpret as the main reason for the spatial uplift patterns observed in the Farsund Basin (Lines 538-540). The Paleogene uplift referred to in Kley (2018), whilst being important at the regional scale and referenced accordingly in the Geological setting, is largely focussed south of the study area and does not concur with our spatial patterns of uplift. We have removed a sentence corresponding to the regional uplift events and have clarified the sentence to make it clear that we are referring to the uplift of the domes to the north and east (Line 533-536).

32. Line 445 – it is obvious that the STZ localises over long time far-field stresses. The crucial question, however, is why there are so many different development histories along-strike of the STZ. Areas with an important Triassic-Jurassic history and others with Lower Cretaceous rifting and as well different degrees of Late Cretaceous deformation. On the other hand most the STZ show an Upper Cretaceous shortening but to different degrees. Maybe the STZ reacts more sensitively to compression as to extension? Maybe the difference and the segmented characteristic is an effect of crustal heterogeneities along the STZ

Response The reviewer raises an interesting point here. At upper crustal depths, the STZ is largely defined as a zone of Late Cretaceous shortening, with this shortening accommodated by a variety of mechanisms (e.g. reverse fault reactivation, long-wavelength folding of basin fill). We propose in this study that, at least at the basin-scale, the prior evolution of the basin plays an important role in how it accommodates these relatively far-field compressional stresses. The presence and prior evolution of the Farsund North Fault controls the structural style of shortening that occurs within the basin. At the more regional scale, we agree with a previous comment raised by the reviewer that the young Early Cretaceous age of the graben, may influence how

it behaves when subject to later compression. In accordance with that comment (addressed in Point 8), we have added some additional text to the revised manuscript stating the relatively young age of the Farsund Basin relative to other structures along the STZ may be more prone to reactivation (Huyghe and Mugnier, 1995) (Line 558-564). During its history, the STZ has largely never been directly subject to either pure compression or extension, so we are unable to comment on whether it reacts more sensitively to one or the other. Instead the STZ typically reactivates via some component of oblique transtension/transpression (Japsen et al., 2007a; Mogensen, 1995). The style of this reactivation is, as the reviewer states, governed by the initial structure and segmented nature of the rift systems along the STZ. We highlight this for the Farsund Basin, and at a more local scale along its northern margin.

33. Line 480-482 – Are there any ideas what the mechanism behind this flexural/monoclinal bulge during the Neogene and the South Scandic Dome is? Dynamic topography? If the STZ will act as a hingeline for uplift during the Neogene facies distributional pattern of the Neogene should show similar trends. Are there any studies on this in regions with a more complete Neogene strat. Column? Your figures do not support the idea of the STZ as a hinge-line for uplift in the Neogene.

Response Based on the regional nature of Paleogene-Neogene uplift, it has been proposed that this relates to a reorganisation of plate tectonic forces and also dynamic topographic effects relating to uplift of the South Scandes and South Swedish domes (Stoker et al., 2005, Japsen et al., 2007a, 2018, Kley et al., 2018). We have added some text to the revised manuscript in the geological history regarding these potential mechanisms (Lines 146-154, 536-537). See also response to Points 2 and 5 above. Because we do not see any preserved Upper Cretaceous-Neogene strata across the study area, we are unable to distinguish individual events and cannot identify any changes in depositional facies. However, based on the bulk uplift values calculated through our well analyses we suggest that the Sorgenfrei-Tornquist Zone represents a "relative" hingeline between areas experiencing large vertical motions to the north,

and smaller vertical motions in the Norwegian-Danish Basin to the south. This is in agreement with previous studies in the area (Japsen et al., 2007a, 2018). We have modified the text to make our interpretation of a "relative" hingeline clearer in the revised manuscript (Line 594-596).

34. Line 490 – Your presented seismic profiles show only the top 6 s TWT of the strat column, what information do the authors have about the geometry and complexity of the fault with depth? A complex fault pattern in the most upper strat column does not have to mean that the fault in the deeper section must have a complex geometry

Response See response to Point 8 above. Along the eastern segment of the fault, the area referred to as "relatively young and geometrically simple" (Line 561), we interpret a simple planar fault geometry at both shallow and deeper levels. We have since clarified this point in the conclusions to highlight that the along-strike changes in fault complexity are most important (Line 605).

35. Line 491-492 – The late cretaceous shortening is the result of far-field stresses. This means that more or less the whole STZ should have been affected by that in a similar way. Therefore, it would be strange if adjacent sections of the STZ would not show similar amounts of shortening or smoothed decreases or increases in a regional trend. In such cases there is a need for local additional effects (strain partitioning, transfer of shortening onto other structures, or a change in the deformation style).

Response This is a good and interesting point raised by the reviewer. The Sorgenfrei-Tornquist Zone represents a buffer to inversion during the Late Cretaceous, and accordingly some shortening is observed along the whole of the structure in some form. Due to the regional and far-field nature of the applied stress, we would not expect any major changes in the amount of shortening experienced at the local scale. However we suggest that this shortening can be accommodated via different mechanisms along-strike, such that whilst the basin may experience changes in the degree of shortening via any one particular mechanism, the regional amount of shortening across the

STZ remains relatively constant. We observe relatively minor Late Cretaceous inversion along the Farsund North Fault; the magnitude of this inversion is similar to that observed along-strike to the west along the Stavnger Fault System in the Egersund Basin (Jackson et al., 2013). We have added some text to this effect in the revised manuscript (Line 517-518).

Figures

Figure 1 – Although a conceptual model, it would be good to highlight the stratigraphy of the region in the sketch

Response We have incorporated some aspects of the Farsund Basin here (namely the colour scheme and unconformity) to enable comparison between this conceptual figure and the observed seismic sections, and also to highlight the difficulties presented in determining the age of the inversion along the fault due to the missing strata at the unconformity. However, this conceptual figure shows the geometry and formation of a typical inversion-related anticline, and is not intended to be specific to any particular lithologies. We believe that incorporating stratigraphic information into this figure will not add value.

Figure 2 Add co-ordinates Altered in revised manuscript

Expand on what is meant by STZ projection This has been changed on the figure to "Along-strike STZ continuation". This is based on projected continuations of the STZ to the west of the Farsund Basin.

Column labels too small in figure 1b. Enlarge section The labels have been enlarged in this figure. In addition, some have been changed to reflect the content of the revised manuscript (i.e. Africa-Iberia-Europe Convergence).

Figure 3 At present-day – Are the faults really of Permo-Carboniferous age? The majority of these faults only offset strata of proposed Carboniferous-Permian Age and show no offset at the base of the Zechstein. Therefore we suggest a likely Carboniferous-

Permian age, in conjunction with previously documented Carboniferous-Permian rift activity in this area (see lines 430). However, we acknowledge that we cannot be certain of this age, and have amended the interpretation to state that the activity is pre-Zechstein, and likely Carboniferous-Permian.

Incomprehensible differences between interpretations of seismic section and the flattened seismic at End Triassic. Why does the thickness of the brown pre-Zechstein unit change? We are uncertain of the thickness of the brown pre-Zechstein unit as we have no direct constraints on the base horizon, which we represent by a dashed line. We have amended our interpretation of the unit to ensure compatibility between the two sections and to ensure that the thicknesses do not drastically change.Àň

Why do faults with offsets on the base pre-Zechstein at the end of Triassic not show offsets in todays picture? These faults have since been modified in the revised version of the figure.

Unusual changes in the distribution and interpretation of Zechstein The interpretation of the Zechstein unit has been reinterpreted across both sections to ensure compatibility. Particular attention has been paid to the location of welds within the Zechstein and the salt structure on the right hand side of the section

Indications for Triassic offset on some faults? Following a re-interpretation of the horizons across the flattened profile, we do not believe there to be any significant Triassic offset along the faults. E-W-oriented extension occurred during the Triassic, as documented in Phillips et al, (2018) and was accommodated along N-S-striking faults. We identify no activity along E-W striking faults during the Triassic, although some may offset the acoustic basement (base upper Permian) horizon, suggesting some relatively minor Permian activity.

Change in extent of Zechstein (i..e depositional limit) This has now been amended in the revised version of the figure.

You have interpreted the Triassic picture in a way that no indications for pre-cursors of the STZ is given. Is this implication meant to be created, that the Farsund Basin coincides with the STZ, but does not show clear pre-Cretaceous pre-cursors? This is intentional and is one of the interesting features of the Farsund Basin, namely that no clear evidence of Carboniferous-Permian extension is present, compared to elsewhere further east along the STZ. Prior to Early Cretaceous rifting, the Farsund Basin was located along the northern margin of the Norwegian-Danish Basin, with only some E-W-directed Triassic extension occurring in the area, producing N-S-striking faults. This figure highlights that, prior to Early Cretaceous rifting, the Farsund Basin and Varnes Graben were continuous and resided along the northern margin of the Norwegian-Danish Basin.

Towards the NDB, you have interpreted a weld, but in the present-day section there is Zechstein again? This has since been rectified in the revised version.

Top Zechstein is not consistent between figures at the large salt structure on the RHS This has also been corrected in the revised version of the manuscript.

Figure 9 – Thinned layers on the central subfigure? A label has been added to the central subfigure to indicate the thinning of strata across the fold. We suggest that this is a consequence of the earlier fault propagation folding that occurred during the Early Cretaceous extension.

Figure 11 – shearing of pre-rift units along blind faults needs thinning of those units or additional minor faulting (in the fault propagation fold. This has now been rectified on the revised figure to show thinning of the blue pre-rift interval across the fault within the fault propagation

Are there indication of decrease of normal thicknesses of pre-rift strata in the hang-ingwall of the main fault? Figure 9b indicates this, 9a and 9c do not. There is no decrease in the thicknesses of the pre-rift strata across the main fault. An additional label has been added to the figure to make this clearer . The step from b to c is not

fully comprehensible. If the onlap geometries and contact relations were as in sketch b before inversion, then inversion of the structure by reactivating the lower cretaceous main fault would result in a different picture. (rotaitons of the contacts of the lower cretaceous onlaps to peeusdo-downlaps. The actual thesis seems plausible, but its graphic implementation is not yet really coherent.

We have redrafted part c of this figure in order to make the geometric relations more apparent. We highlight that the onlapping geometries of the Lower Cretaceous strata would be rotated to form pseudo-downlaps and also highlight the variable fold amplitude with depth.

Please also note the supplement to this comment:
https://se.copernicus.org/preprints/se-2020-27/se-2020-27-AC2-supplement.pdf

[Figure]

**Fig. 1.** Revised Fig 2

[Figure]

[Figure]

**Fig. 2.** Revised Fig 3

**a)**

**b)**

**c)**

TWT (s)

1 km

L. Cret.

Jurassic

Triassic

Carb/Perm.

Section restorable
pre-Cretaceous

1 km

Upwards decrease in
fold amplitude

Truncation
of fold strata

Antithetic crestal
faulting

Inversion-related
anticline

U. Jurassic delta

Truncation of L. Cret. strata

Onlapping onto fold

Minor crestal
faulting

Maximum fold amplitude in
lowermost Cretaceous

Onlapping onto fold limb

Thinning of strata

Truncation of
fold crest

Thinning of strata
on fold limb

Tight inversion fold

Farsund Dyke
Swarm

**Fig. 3.** Revised Fig 9

[Figure]

**Fig. 4.** Revised Fig 11

[Figure]

**Fig. 5.** Revised Fig 5

**Supplement:**

[revised manuscript text omitted]